# Enhancing Deep Partial Label Learning via Casting it to a Satisfiability Problem

## Abstract

Partial label learning (PLL) is a challenging real-world problem in the field of weakly supervised learning, in which each data instance contains a set of candidate labels with multiple ambiguous labels and one gold label. Although recent progress in PLL using deep representation learning has led to significant advances, the methods continue to experience significant performance drop on data with high label ambiguity and fine-grained categories. By *casting PLL into a satisfiability problem* and incorporating a loss based on this reduction, we show that the accuracy of those techniques can be further improved. We establish several key theoretical properties of the proposed SATisfiability-based (SAT) loss and its learning error bound. Our extensive empirical comparison reveals that the proposed loss improves over existing PLL techniques by up to $25.12\%$ on multi-class benchmarks and $12.50\%$ on fine-grained categorized benchmarks.

## 1 Introduction

Traditional supervised learning models rely on precise, unambiguous labels. However, common labeling methods, including crowd-sourced annotations, where the expertise of annotators varies significantly, and unsupervised learning models, which often struggle with ambiguous or challenging cases, can have difficulty giving clear-cut and definitive labels. These practical concerns make *partial label learning* (PLL) (Cour et al., 2011b; Tian et al., 2023) a necessity rather than a choice, especially in situa-

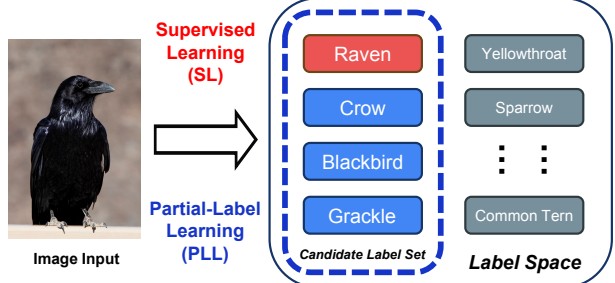

**Image Input**

Figure 1: An image of a raven may be difficult to distinguish within the set of candidate labels.

tions where obtaining perfectly labeled data is impractical or impossible. *Label disambiguation* is the core challenge of PLL where training models need to effectively handle and learn from data in which each instance is associated with a set of candidate labels, only one of which is correct. As shown in Figure 1, unless a bird expert, a human annotator would have a hard time distinguishing the "Raven" from "Crow", "Blackbird" or "Grackle".

Recent deep neural network (DNN) approaches in PLL have made real practical progress. Two of the most notable works, PICO (Wang et al., 2022b) (or PICO$^+$ (Wang et al., 2022c)) and PaPi (Xia et al., 2023b), use prototype-based representation learning to guide label disambiguation in the training process. These studies have demonstrated remarkable performance on standard datasets such as CIFAR-10 (Krizhevsky, 2009), MNIST(LeCun et al., 1998), SVHN(Netzer et al., 2011).

Relying on the assumption that points closer in the feature space are more likely to share the same gold label, PICO and PICO$^+$ proceed by learning prototypes for each class in a contrastive manner. They generate pseudo labels for each data instance based on their normalized similarity with prototype

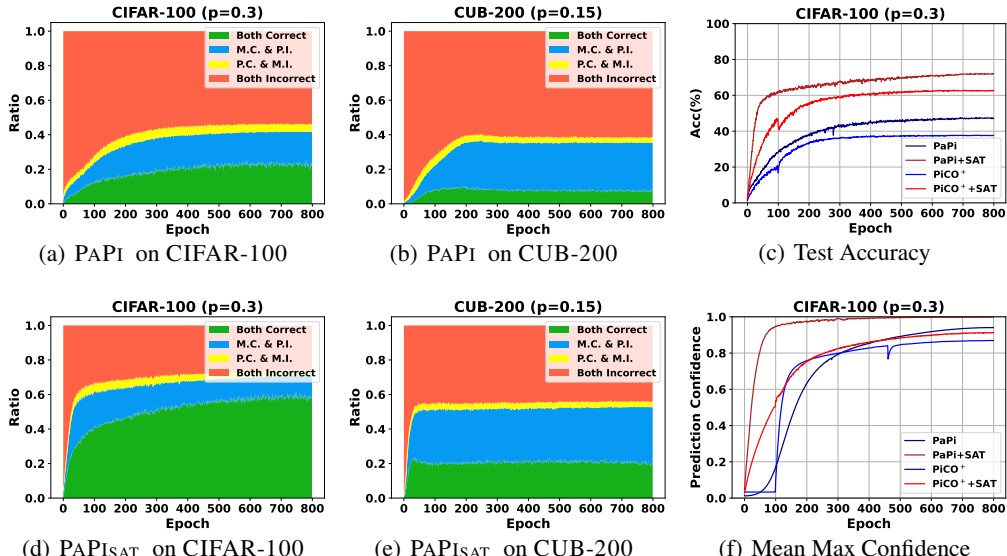

Figure 2: Comparison of prediction accuracy and confidence between PAPI and PICO$^+$ with their integration of SAT loss. (a) and (b) show the classification percentages by Both Correct, Classifier-only, Prototypes-only, and Both Incorrect of PAPI on the test data from CIFAR-100 and CUB-200. (d) and (e) show the percentage changes by SAT-enhancement, i.e. PAPI$_{SAT}$, on the same test data. (c) and (f) shows the changes of test accuracy and average prediction confidence of PICO$^+$ and PAPI with their SAT-counterparts during the training process.

embeddings. Then, the cross entropy loss is adopted to guide label disambiguation using the pseudo label guidance. However, interleaving the process of label disambiguation with prototype learning may lead to undesired phenomena. In particular, erroneous label disambiguation may negatively affect representation learning by trying to reduce the distance between embeddings of instances that belong to different classes, which may consequently introduce further errors for following training. Such an observation led to PAPI (Xia et al., 2023a).

Despite its improvements over PICO, PAPI is still prone to undesired interactions between the label disambiguation and the representation learning processes, in scenarios in which the candidate sets of labels are large or when data from extremely similar categories (Wah et al., 2011; Nilsback & Zisserman, 2008). Figures 2(a) and 2(b) show for each training epoch in PAPI, the percentage of test images that (i) were correctly classified by the classifier and incorrectly classified based on the prototypes (blue area), (ii) were incorrectly classified by the classifier and correctly classified based on the prototypes (yellow area), (iii) were correctly classified both under the classifier and based on the prototypes (green area), and finally, (iv) were incorrectly classified under both cases (orange area). In Figure 2(a) the PLL dataset is constructed out of CIFAR-100, while in Figure 2(b) the PLL dataset is constructed out of CUB-200. In both cases, each training sample has roughly 30 candidate labels. As illustrated by the yellow areas of Figure 2(a) and 2(b), the percentage of images correctly classified based on the prototypes and incorrectly classified by the classifier is minimal relative to the entire dataset. Furthermore, under CUB-200, the percentage of images in the green area shows a decreasing trend, see Figure 2(b). This suggests that the guidance from prototypes is not reliable, and can even be counterproductive in realistic scenarios where data categories are very similar.

*What is the root cause of this undesired behavior?* Both PICO (and its successor PICO$^+$) and PAPI put emphasis on prototype learning, simply using the cross entropy of the pseudo labels as a classification loss term. However, choosing the cross entropy loss on highly noisy pseudo labels may have catastrophic consequences: it is already known that the cross entropy loss is not robust to noise (Feng et al., 2021). To tackle noisy labels, an approach would be to employ loss functions that are tolerant to noisy labels (Wang et al., 2022c). However, all these techniques work under certain assumptions on how noise is modeled. These assumptions are violated in the standard PLL setting: we only know that the gold label is in the set of candidate labels, with no other information.

To overcome the limitations of PICO and PAPI, and motivated by the observations that the classifier was always more correct than prototype-based classification at the beginning and sometimes this advantage was even maintained until convergence, we remove PICO's and PAPI's cross entropy-based classification loss term and incorporate techniques for training neural classifiers subject to logical constraints (Wang et al., 2023). In particular, we show that the PLL constraints can be naturally encoded into formulas in propositional logic. These formulas are subsequently used to compute losses capturing the degree to which the classifiers' scores abide by them, hereby transforming PLL into a satisfiability problem. We define this logic-based formulated loss as *SATisfiability-based (SAT) loss* and provide theoretical justification for its effectiveness on difficulty scenarios as high label ambiguity and fine-grained candidate categories. Further, we empirically demonstrate that satisfiability-based training can serve as a standalone loss function or enhance the state-of-the-art PLL methods beyond PICO$^+$ and PAPI via seamless integration. As shown in Figure 2(c) and (f), integrating SAT loss effectively enhanced the classification (Test Accuracy) and label disambiguation performance (Mean Max Confidence) of PICO$^+$and PAPI . Based on the comparison between Figure 2(a)(b) and (d)(e), it also improved the consistency between the classifier and the prototype predictions as a better classifier generates high-quality representations. We summarize our **main contributions** as below:

**Methodology.** We are the first to thoroughly assess satisfiability-based learning in the context of PLL and its merits when integrated with state-of-the-art PLL techniques focusing on learning better representations (Wang et al., 2022b;c; Xia et al., 2023a; Wu et al., 2022a). The obtained empirical results are state-of-the-art in PLL. The proofs of all formal statements, as well as more details on our empirical analysis are in the appendix.

**Theory.** Beyond empirically assessing satisfiability-based PLL, we theoretically analyze the properties of the proposed method. We prove several natural properties of satisfiability-based PLL: that the loss is non-increasing the larger candidate label sets become, see Corollary 5.2, and that it favors low entropy classifiers, Proposition 5.3. In addition, we provide the Rademacher-style error bound, showing the learnability and generalization ability of SAT loss, see Theorem 5.4.

**Experiments.** We validate the performance of logic-inspired PLL training empirically, via the most thorough, to our knowledge, experimental evaluation. Our assessment shows that satisfiability-based training leads to consistently higher classification accuracy over all settings compared with previous state-of-the-art techniques. Its integration brings up to $25.12\%$ increase on CIFAR-100 for PICO$^+$ and $12.50\%$ improvement on CUB-200 for PAPI.

## 2 RELATED WORK

The literature on PLL is very rich. We quickly summarize the most well-known theoretical results. The *small ambiguity degree* assumption (Cour et al., 2011a; Cabannes et al., 2020) has been central to showing several formal properties including learnability (Liu & Dietterich, 2014) and classifier consistency (Lv et al., 2020; Cabannes et al., 2020; Wang et al., 2022a). Regarding the latter, the authors in (Lv et al., 2020) show classifier consistency via training under the *minimal loss*, while the authors in (Cabannes et al., 2020) show classifier consistency via the *infimum loss*. Both the minimal and the infimum losses can be seen as a special case of *Semantic Loss* (SL) (Xu et al., 2018b). In addition, similarly to (Lv et al., 2020; Cabannes et al., 2020), we prove classifier consistency and provide Rademacher-style error bounds, see Section 5.

*Risk-consistent* PLL techniques have been also proposed (Feng et al., 2020; Wen et al., 2021), where a PLL technique is risk-consistent (Xia et al., 2019) if by increasing the size of PLL samples from $P(X, S)$, the empirical risk based on those samples converges to the expected risk based on fully-supervised supervised samples from $P(X, Y)$. Classifier- and risk-consistent techniques were proposed in (Feng et al., 2020), named CC and RC, respectively, based on the assumption that the PLL training samples abide by a specific data generation process. This research was later extended by (Wu & Sugiyama, 2021). The authors in (Wen et al., 2021) introduced a family of loss functions named *leveraged weighted loss* (LWC) and proved risk and Bayes consistency under a data generation process that extends the one from (Feng et al., 2020). Differently from (Feng et al., 2020; Wen et al.,

2021), our work makes no assumption on the data generation, while the empirical results demonstrate superior classification accuracy against CC, RC, and LWC, see Section 6.

Advancements in deep learning have further expanded the PLL exploration (Xu et al., 2021; Wang et al., 2022b;c; Wu et al., 2022a; Xia et al., 2023a). The authors in (Xu et al., 2021) introduced VALEN, a technique based on variational inference, for PLL settings in which the candidate labels in $s$ are dependent on the features of $x$. The key idea is to associate each $x$ with a latent label distribution over the labels in $s$, representing the probability to which each $l \in s$ describes the features of $x$. Recently, (Wang et al., 2022b;c) adopted contrastive representation learning (Chen et al., 2020) for PLL, significantly improving the state-of-the-art results on several image classification benchmarks. Despite its remarkable performance, the authors in (Xia et al., 2023a) observed that PiCO's contrastive learning component and its label disambiguation process that is based on the class embedding prototypes may negatively affect the training process. Motivated by these finds, the authors proposed PaPi (Xia et al., 2023a). PaPi gets rid of contrastive learning and relies on aligning the distribution over the pseudo-labels of $x$ with a distribution over the distances of $x$'s feature embedding to the representative feature embeddings of each class. In addition, PaPi relies on the classifier's predictions for label disambiguation and not to the class prototypes. Another recently proposed PLL technique that relies deep learning is DPLL (Wu et al., 2022a). Both PiCO and PaPi rely on the cross-entropy loss for label disambiguation. Our work instead shows that via replacing this loss term with $\ell^{SAT}$, we can have major improvements in the accuracy of the learned classifiers, see Section 6.

For completeness, we conclude with work on other learning settings. One of the closest fields is neuro-symbolic learning, whose key focus is the integration of deep learning with logical reasoning. Works as Tsamoura et al. (2021); Buffelli & Tsamoura (2022) explore frameworks that combine neural network outputs with symbolic reasoning processes, enhancing the system's ability to perform tasks requiring complex reasoning or knowledge representation. Despite its promise, neuro-symbolic learning faces challenges, particularly in seamlessly integrating neural and symbolic components and scaling these systems for complex, real-world applications. There are many existing works (Xu et al., 2018a; Manhaeve et al., 2018; Wang et al., 2019; Huang et al., 2021) explored utilizing logic constraints to train neural networks. A few of the close works deployed logic constraints into multi-instance partial label learning learning (Zombori et al., 2023), which are different from what we emphasized. In this work, we aim to demonstrate that training neural network with logic constraints can enhance their robustness in handling label ambiguity under standard partial label learning.

Complementary to ours is the work in (Ferber et al., 2019; Vlastelica et al., 2019; Paulus et al., 2021) that integrates combinatorial solvers into deep models. The integration of combinatorial optimization methods, as explored in these works, provides a direct link to PLL under logic constraints. Specifically, decision-focused learning methods like MIPaaL (Ferber et al., 2019) and CombOptNet (Paulus et al., 2021) align well with PLL, where selecting the correct label from a set of candidate labels can be framed as an optimization problem. The incorporation of integer programming and combinatorial solvers as layers in deep networks, as demonstrated in these papers, suggests that similar approaches could be applied to PLL, especially under the scenario where candidate label sets need to be disambiguated efficiently.

## 3 PRELIMINARIES

In this section, we introduce the notation and the formulation of supervised and partial label learning.

**Supervised Learning.** Let $\mathcal{X}$ denote the input space and $\mathcal{Y} = [C]$ denote a finite label space, where $[C] = \{1, \ldots, C\}$. For an annotated dataset $\mathcal{D}_N = \{(x_i, y_i)_{i=1}^{N}\}$ sampled from an unknown distribution $P(X, Y) \in \mathcal{X} \times \mathcal{Y}$, each data instance $x_i \in \mathcal{X}$ has a label $y_i \in \mathcal{Y}$. A distribution $P(X, Y)$ is *deterministic* if $P(Y = y | X = x) = 0$ unless $y$ is the gold label $y^*$ of $x$. We define the problem of *learning* as the process of finding the scoring function $f$ from the hypothesis space $\mathcal{F}$ that can minimize the classification risk over the given training samples $\mathcal{D}_N$. We define the classification risk

subject to a given loss function $\ell$ as:

$$\mathcal{R}(f; \ell) := \mathbb{E}_{(x,y) \sim P(X,Y)} [\ell(f(x), y)] \tag{1}$$

The lower $\mathcal{R}(f; \ell)$ becomes, the better $f$ fits the data. For optimal classifier $f^*(\cdot)$, we have $\mathcal{R}(f^*; \ell) = 0$. We use $\widehat{\mathcal{R}}(f; \ell; \mathcal{D}_N)$ to denote the empirical risk, i.e. the average risk over $\mathcal{D}_N$.

**Partial label learning.** Differently from supervised learning where each input is paired with a gold label $y_i^*$, in *partial label learning* (PLL), each data instance $x_i$ is associated with a candidate label set $s_i$, which contains one gold label and a set of candidate labels, i.e. $s_i = \{y_i^*\} \cup z_i$, where $z_i$ is the additional label set. In particular, partial labeled data is drawn from a *partial distribution* $P(X, S)$ over $\mathcal{X} \times \mathcal{S}$, where $\mathcal{S} \subset 2^{\mathcal{Y}}$ is the set of all subsets of $2^{\mathcal{Y}}$ excluding $\mathcal{Y}$ and the empty set, that is *compatible* with $P(X, Y)$ (Cabannes et al., 2020), i.e., there exists a probability measure $P(X, Y, S)$ over $\mathcal{X} \times \mathcal{Y} \times \mathcal{S}$, such that $P(X, Y)$ is the marginal of $P(X, Y, S)$ over $\mathcal{X} \times \mathcal{Y}$, $P(X, S)$ is the marginal of $P(X, Y, S)$ over $\mathcal{X} \times \mathcal{S}$, and $P(S = s | Y = y) = 0$ if $y \notin s$. In analogy to supervised learning, the objective is to find a classifier $f$ that minimizes a *partial risk* subject to a *partial loss function* $\ell : \mathbb{R}^c \times \mathcal{S} \to \mathbb{R}^+$: $\mathcal{R}(f; \ell) := \mathbb{E}_{(x,s) \sim P(X,S)}[\ell(f(x), s)]$

**Classifiers.** We consider *scoring functions* of the form $f : \mathcal{X} \to \Delta_C$, where $\Delta_C$ is the space of probability distributions on $\mathcal{Y}$ (e.g., $f$ outputs the softmax probabilities of a neural network). We use $f^j(x)$ to denote the $j$-th output of $f(x)$. A scoring function $f$ induces a *classifier* whose *prediction* on $x$ is defined by $[f](x) := \arg\max_{j \in [C]} f^j(x)$. We use $\mathcal{F}$ and $[\mathcal{F}]$ to denote the space of scoring functions and the space of classifiers induced by $\mathcal{F}$, respectively. We use the Rademacher complexity to characterize the complexity of hypothesis space $\mathcal{F}$ (Shalev-Shwartz & Ben-David, 2014), which is formally defined in Section A.1.

**Loss function.** We present several commonly used loss functions to facilitate the discussion in the following sections. *zero-one loss* defined as $\ell^{01}(f(x), y) := \mathbb{1}\{f(x) = y\}$ is commonly used in supervised multi-class learning task. We also consider the *partial zero-one loss* (Cour et al., 2011b), $\ell_P^{01}(f(x), s) := \mathbb{1}\{[f](x) \in s\}$ for partial label setting. *Cross entropy loss* is frequently used for label disambiguation, which defined as $\ell^{ce}(f(x), e^y) := -\sum_{j \in [c]} e^y[j] f^j(x)$, where $e^y[j]$ denotes the $j$-th entry of the one-hot vector $e^y$. Different partial loss functions have been proposed in the literature, including the *infimum loss* (Cabannes et al., 2020) and a *modified negative log likelihood loss* (NLL) (Wu et al., 2022a) given by $\ell^{nll}(f(x), s) := -\sum_{j \in \mathcal{Y} \setminus s} \log(1 - f^j(x))$. We refer to the risk of $f$ subject to the above losses as the zero-one, partial zero-one, cross-entropy and NLL risk and denote them by $\mathcal{R}^{01}(f)$, $\mathcal{R}_P^{01}(f)$, $\mathcal{R}^{CE}(f)$, and $\mathcal{R}^{nll}(f)$ respectively.

## 4 METHODOLOGY

### 4.1 PLL AS A SATISFIABILITY PROBLEM

We overcome the limitations of training using pseudo labels and the cross entropy loss by reducing PLL to a satisfiability problem. This essentially requires encoding the set of candidate partial labels $s$ of each training sample $(x, s)$ from $P(X, S)$ as a Boolean formula $\varphi_s$ that is true if and only if the Boolean variable associated with some label in $s$ is true. This Boolean formula is the conjunction between two formulas:

$$\varphi_s := \bigvee_{j \in s} X_j \wedge \bigwedge_{j, j' \in s : j \neq j'} \neg(X_j \wedge X_{j'}) \tag{2}$$

We use a slight abuse of notation to denote $X_j$ as the Boolean variable that is uniquely associated to label $j \in s$; $\vee$ (OR) and $\wedge$ (AND) are the logical connectives. The probability that $X_j$ becomes true subject to classifier $f$ and sample $x$ is $f^j(x)$. We provide an example below that illustrates how we can formulate the logical sentence constraints derived from partial labels.

**Example 4.1.** *Given a dataset $D_s$ with label space $\mathcal{Y} = \{bird, airplane, tiger, dog, lion\}$, we have a data instance $x$ and its candidate label set $s = \{tiger, dog, lion\}$, meaning that the first three labels*

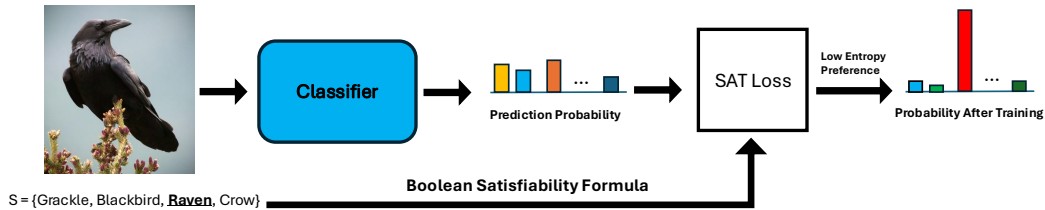

Figure 3: Learning with SAT loss

*("tiger", "dog", "lion") are candidate labels. To encode this as a Boolean formula, we will use the Boolean variables $X_{label}$ representing the labels. The formula $\varphi_s$ is constructed in two parts:*

*The **disjunction** over the candidate labels in $s$, i.e., at least one of the candidate labels must be true:*

$$\bigvee_{j \in s} X_j = X_{tiger} \vee X_{dog} \vee X_{lion}$$

*This ensures that one of the labels "tiger", "dog", or "lion" must be true. The **conjunction** over all pairs of candidate labels to ensure that no two labels from $s$ are true simultaneously:*

$$\bigwedge_{j,j' \in s: j \neq j'} \neg(X_j \wedge X_{j'}) = \neg(X_{tiger} \wedge X_{dog}) \wedge \neg(X_{tiger} \wedge X_{lion}) \wedge \neg(X_{dog} \wedge X_{lion})$$

*This ensures that only one of the labels "tiger", "dog", or "lion" can be true. Combining these two parts, the final Boolean formula $\varphi_s$ is:*

$$\varphi_s := (X_{tiger} \vee X_{dog} \vee X_{lion}) \wedge \neg(X_{tiger} \wedge X_{dog}) \wedge \neg(X_{tiger} \wedge X_{lion}) \wedge \neg(X_{dog} \wedge X_{lion})$$

*How do we compute a loss out of this formula?* Assuming that each $X_j$, for $j \in s$, becomes true with probability $f^j(x)$, a natural loss should penalize formulas with low probability of being satisfied and reward formulas otherwise. By omitting the right part of the conjunction, i.e., if we consider only the subformula $\bigvee_{j \in s} X_j$ from equation 2 (disjunction), and using previous results from probabilistic databases (Suciu et al., 2011), this probability becomes $1 - \prod_{j \in s}(1 - f^j(x))$, which is simply one minus the probability all $X_j$ being simultaneously false. To satisfy the properties mentioned at the very beginning of this paragraph, the resulting SATisfiability-based (SAT) loss takes the form:

$$\ell^{SAT}(f(x), s) := -\log(1 - \prod_{j \in s}(1 - f^j(x))) \tag{3}$$

**Remark 4.2.** *The definition of $\ell^{SAT}$ does not take into account the right-hand side (conjunction) of (2), that is the mutual exclusiveness constraints. Disregarding those constraints is harmless in our case, as the softmax scores of $f$ jointly with the optimization objective (that is risk minimization, see Section 5) implicitly enforce mutual exclusiveness.*

**Remark 4.3.** *We refer to the quantity $1 - \prod_{j \in s}(1 - f^j(x))$ as the probability of formula $\bigvee_{j \in s} X_j$, aligning with (Suciu et al., 2011; Chavira & Darwiche, 2008). However, the probability of that formula coincides with its* product t-norm *from the fuzzy logic literature (Diligenti et al., 2017), by rewriting it via De Morgan law to $\neg(\bigwedge_{j \in s} \neg X_j)$ (see Section 3 from (Diligenti et al., 2017)).*

## 5 THEORETICAL ANALYSIS

In this section, we provide a theoretical justification to show why the proposed SAT loss can benefit label disambiguation under challenging scenarios such as high label ambiguity and fine-grained categories. The primary challenge brought by high label ambiguity is that each data instance will have a larger set of candidate labels. On the other hand, compared to standard datasets like CIFAR-10, fine-grained categorization data involves data that comes from the same super-category, such as

different species of birds. While this is more reflective of real-world scenarios, it also leads to the model assigning similar prediction probabilities to multiple candidate classes, making it difficult to distinguish between them.

To quantify the label ambiguity level in a given dataset $\mathcal{D}_N$, we adopt the concept of *small ambiguity degree*, which was widely used in PLL literature (Liu & Dietterich, 2014; Lv et al., 2020; Cabannes et al., 2020; Wang et al., 2022a).

**Definition 5.1** (Small Ambiguity Degree). *The small ambiguity degree $\gamma$ is a measure of the hardness of the learning under partial label supervision. Given distribution $P(X, Y, S)$, where $X, Y, S$ is the variables of input, label and candidate label set, we have the definition:*

$$\gamma := \sup_{(x,y^*)\in\mathcal{X}\times\mathcal{Y}, P(x,y^*)>0, j\in s\backslash y^*} P(j \in s | X = x, Y = y^*) \tag{4}$$

Here $j$ denotes a non-true candidate label or additional label. A partial labeled instance satisfies the small ambiguity degree condition when $0 \leq \gamma < 1$, signifying that no co-occurring labels are present alongside the ground truth. Next, we present an extensive analysis of learning via minimizing $\ell^{SAT}$ to show the properties of SAT loss can mitigate these challenges.

## 5.1 Robustness to high label ambiguity.

As shown in the Corollary 5.2, we first discuss that the SAT loss holds the property that its risk is non-increasing as the small ambiguity degree increases.

**Corollary 5.2** (Monotonicity under ambiguity). *Let $P_1(X, S)$ and $P_2(X, S)$ be two partial distributions over $\mathcal{X} \times \mathcal{S}$ with ambiguity degrees $\gamma_1$ and $\gamma_2$, such that for each sample $(x, s_1)$ occurring with probability $\alpha$ in $P_1(X, S)$, a sample of the form $(x, s_2)$ occurs with probability $\alpha$ in $P_2(X, S)$ and $s_1 \subseteq s_2$, and hence, $\gamma_1 \leq \gamma_2$. Then, $\mathbb{E}_{(x,s)\sim P_1(X,S)}[\ell^{SAT}(f(x), s_1)] \geq \mathbb{E}_{(x,s)\sim P_2(X,S)}[\ell^{SAT}(f(x), s_2)]$.*

If $s_1 \subseteq s_2$, and hence $\gamma_1 \leq \gamma_2$ then $1 - \prod_{j\in s_1}(1 - f^j(x)) \geq 1 - \prod_{j\in s_2}(1 - f^j(x))$. Taking the expectation considering samples from $P_1(X, S)$ and $P_2(X, S)$, respectively, leads to the above corollary. This property indicates that the risk of SAT loss is determined by the model prediction and will not be significantly affected by high label ambiguity as NLL losses, such as *partial cross entropy loss* and *modified negative log likelihood loss*(Wu et al., 2022b). As an example, the *partial cross entropy loss*, defined as $\ell_P^{CE} = -\sum_{j\in s} \log(1 - f^j(x))$, will result in a higher loss value as the size of candidate label set increasing.

## 5.2 Label Disambiguation via Entropy Minimization

Fine-grained categorized data (Wah et al., 2011; Nilsback & Zisserman, 2008) is extremely challenging for previous PLL methods (Wang et al., 2022b;c; Xia et al., 2023a) using *cross entropy loss* for label disambiguation, as the high similarity among the candidate labels causing the pseudo labels generated by these methods become noisy and unreliable.

The advantage of SAT loss on fine-grained categorization data can be attributed to its two key characteristics. On the one hand, SAT loss gets rid of the pseudo label, which may lead to erroneous guidance by the representation learning module or self-supervised manner. On the other hand, SAT loss forces the classifier to concentrate the prediction probability on one single candidate label during the training process. To show that, we provide the second important property of SAT loss below, the *low entropy preference*.

**Proposition 5.3.** *[Low entropy preference] Let $H(f(x); s) := -\sum_{j\in s} f^j(x) \log f^j(x)$ denote the entropy of the PLL sample $(x, s)$ subject to $f$. Then $\ell^{SAT}(f(x), s) \propto H(f(x); s)$.*

Notice that favoring low-entropy solutions, essentially means that $\ell^{SAT}$ penalizes classifiers that output uniform (or close to uniform) softmax probabilities, as visualized in Figure 3. When the

prediction entropy $H(f(x), s)$ is low, indicating the model is very confident in a single class and gives it a high prediction probability, encouraging the label disambiguation process. We provide the proof of the above proposition in Appendix A.3.

### 5.3 ERROR BOUND WITH SAT LOSS

We study the error bound between the SAT loss and the *zero-one loss*. As the risk minimization under the *zero-one loss* is impractical even for a linear classifier, therefore we use SAT loss as the surrogate loss. We aim to show that for a classifier $f$ trained on a partial labeled dataset $\mathcal{D}_N$, as $N$ goes to infinity, then the partial risk of SAT loss subject to $f$ converges to the risk of *zero-one loss*.

We discuss the error bound of SAT loss for a partial label learning dataset $\mathcal{D}_N$ with size $N$ under a small ambiguity degree $\gamma$. We extend the small ambiguity degree to the uniformed partial label learning scenario that $\forall j \in s$, we have $P(j \in z | X = x, Y = y^*) = \gamma$, which means given a partial label learning instance $x$, any non-true labels have a probability of $\gamma$ to be in the candidate label set $s$. The average candidate label number can be calculated as $|s| = \lfloor (C-1)\gamma + 1 \rfloor$. We show the error bound below and provide full proof in Appendix A.4.

**Theorem 5.4.** *[Error bound under small ambiguity degree] Given partial labeled dataset $\mathcal{D}_N$ with small ambiguity degree $\gamma \in (0, 1)$, $\forall \epsilon, \delta \in (0, 1)$, with probability at least $1 - \delta$, we have:*

$$\mathcal{R}^{01}(f; \mathcal{D}_N) \leq \frac{1}{1-\gamma} \left( \widehat{\mathcal{R}}^{SAT}(f; \mathcal{D}_N) + 2\sqrt{(C-1)\gamma + 1}\,\Re_N(\mathcal{F}) + \sqrt{\frac{log(1/\delta)}{2N}} \right). \quad (5)$$

$\widehat{\mathcal{R}}^{SAT}(f; \mathcal{D}_N)$ *is the empirical risk of SAT loss $\ell^{SAT}$ over an arbitrary partial labeled dataset $\mathcal{D}_N$; $\Re(\mathcal{F})$ denotes the Rademacher complexity (Definition A.1) of hypothesis space $\mathcal{F}$.*

*Proof Sketch.* To prove Theorem 5.4, we first utilize the inequality between *zero-one loss* and *partial zero-one loss* (Proposition A.4). Then we show the Lipschtness of the SAT loss (Lemma A.3). We show that SAT loss is upper bound by *partial cross-entropy* loss and lower bound by partial zero-one loss (Lemma A.5). We get the final result by Applying standard Rademacher complexity bounds.

**Remark 5.5.** *The error bound we get in Theorem 5.4 indicates that the increasing of small ambiguity degree,i.e. $\gamma$ or number of class $C$ in the training data will increase the learning difficulty, which is intuitive. In approach to the optimal classifier, the training set is expected to satisfy the quantity condition $N \to \infty$, we have the empirical risk of SAT loss $\widehat{\mathcal{R}}^{SAT}(f; \mathcal{D}_N)$ approaches to the risk of ideal zero-one loss, which shows the learnability and generalization ability of SAT loss.*

## 6 EXPERIMENTS

**Datasets.** We consider a variety of multi-class benchmarks, CIFAR-100 (Krizhevsky, 2009) (100 classes) and TINY IMAGENET (Chrabaszcz et al., 2017) (200 classes) and fine-grained classification datasets, CUB-200 (Wah et al., 2011) (200 classes), and OXFORD FLOWER 102 (Nilsback & Zisserman, 2008) (102 classes). To generate PLL scenarios out of the above benchmarks, we adopted the steps from Wang et al. (2022b;c); Xia et al. (2023a). In particular, we generated conventional uniform partial label setting by flipping negative labels $y' \neq y$ to false positive labels with a probability $q = P(y' \in s | y' \neq y)$, all $C-1$ labels that are different from the gold one have a uniform probability to be false positive. To form the candidate label sets, the gold label is aggregated to the flipped ones. The above means that each candidate label set has in total $q \times c$ labels on average. We refer to $q$ as the *partial label rate*. We considered $q \in \{0.1, 0.2, 0.3\}$ for CIFAR-100, TINY IMAGENET, and OXFORD FLOWER 102, and $q \in \{0.05, 0.1, 0.15\}$ for CUB-200. Finally, we carried out experiments in instance-dependent partial label settings (Xu et al., 2021) using CIFAR-10 and CIFAR-100. In this setting, we adopted the steps in (Xu et al., 2021) to generate PLL scenarios and, in particular, by utilizing the pre-trained model's predictions as the label flipping probabilities.

**Baselines.** We compared the efficiency of $\ell^{SAT}$ against state-of-the-art classifier- and risk-consistent PLL techniques that do not rely on representation learning, namely LWC (Wen et al., 2021), RC (Feng

Table 1: Mean classification accuracy on CIFAR-100, TINY IMAGENET, CUB-200, and OXFORD FLOWER 102 for uniform partial labels. **Best results** across all baselines are in red.

| METHODS | CIFAR-100 | | | TINY IMAGENET | | | OXFORD FLOWER 102 | | | CUB-200 | | |
|---|---|---|---|---|---|---|---|---|---|---|---|---|
| | $q=0.1$ | $q=0.2$ | $q=0.3$ | $q=0.1$ | $q=0.2$ | $q=0.3$ | $q=0.1$ | $q=0.2$ | $q=0.3$ | $q=0.05$ | $q=0.1$ | $q=0.15$ |
| CC | 67.19% | 63.55% | 44.32% | 52.03% | 31.28% | 12.63% | 66.43% | 33.66% | 21.54% | 48.86% | 22.45% | 12.32% |
| RC | 66.36% | 56.57% | 33.71% | 34.16% | 7.82% | 2.74% | 84.55% | 52.90% | 39.79% | 59.18% | 27.82% | 18.08% |
| LWC | 67.60% | 60.31% | 34.18% | 32.50% | 8.19% | 3.46% | 83.18% | 52.41% | 44.12% | 56.85% | 28.03% | 18.85% |
| SAT | 71.05% | 70.31% | 51.74% | 54.10% | 50.17% | 29.02% | 92.16% | 81.36% | 61.26% | 69.75% | 58.85% | 37.06% |
| PAPI | 81.14% | 79.77% | 47.55% | 52.44% | 34.29% | 22.34% | 95.32% | 92.11% | 82.83% | 70.40% | 53.31% | 37.05% |
| PAPI$_{SAT}$ | 81.16% | 80.11% | 71.56% | 58.83% | 52.96% | 44.26% | 95.34% | 93.58% | 85.01% | 72.16% | 61.52% | 49.55% |
| PICO$^+$ | 68.20% | 57.15% | 37.76% | 39.89% | 17.86% | 7.74% | 88.99% | 85.55% | 75.86% | 66.59% | 57.37% | 42.45% |
| PICO$^+$$_{SAT}$ | 73.44% | 67.44% | 62.88% | 51.21% | 46.22% | 35.16% | 92.30% | 86.67% | 78.22% | 69.26% | 62.18% | 51.28% |
| DPLL | 78.24% | 75.67% | 63.91% | 52.13% | 20.47% | 6.74% | 91.33% | 67.52% | 42.17% | 67.36% | 38.63% | 21.73% |
| DPLL$_{SAT}$ | 78.27% | 76.28% | 66.49% | 55.55% | 35.77% | 17.16% | 92.16% | 72.17% | 46.21% | 70.33% | 42.16% | 27.28% |

et al., 2020), CC (Feng et al., 2020). We use SAT to denote the cases where training is conducted via minimizing the $\ell^{SAT}$ loss only. We also compared the classification accuracy of PICO$^+$ (Wang et al., 2022c) and PAPI (Xia et al., 2023a) against their $\ell^{SAT}$-based counterparties PICO$^+$$_{SAT}$ and PICO$^+$$_{SAT}$, by replacing the *cross entropy loss* with $\ell^{SAT}$. For DPLL (Wu et al., 2022a), we added $\ell^{SAT}$ as a new term in the loss. We refer to this variant as DPLL$_{SAT}$.

**Implementation.** We followed the official implementation for all baselines (Wang et al., 2022c; Xia et al., 2023a; Wu et al., 2022a; Wen et al., 2021; Feng et al., 2020). In all experiments, we adopted a ResNet18 classifier and ran training for 800 epochs in each case. The learning rate was set to 0.01 with weighted decay. Similarly to (Wang et al., 2022c) and (Xia et al., 2023a), we used a pre-trained ResNet18 as the initial parameters for experiments on CUB-200 and OXFORD FLOWER 102 and randomly initialized for the others. The experiments ran on 8 NVIDIA RTX A6000 GPUs.

### 6.1 EMPIRICAL RESULT

We provide experiment result analysis in this section. Specifically, Table 1 shows results for uniform partial labels on CIFAR-100 and TINY IMAGENET and fine-grained benchmarks OXFORD FLOWER 102 and CUB-200. Table 2 shows results for instance-dependent partial labels on CIFAR-10 and CIFAR-100. We provide further details and experimental results on noisy partial label setting and real-world benchmarks as PASCAL VOC (Everingham et al.) in the Appendix B.

**Results on Multi-class Datasets.** Table 1 shows that the integration of $\ell^{SAT}$ notably enhances the classification accuracy of the baseline method across all scenarios on CIFAR-100 and TINY IMAGENET. Specifically, on CIFAR-100 and when $q=0.3$, PAPI$_{SAT}$ has accuracy 71.56% compared to the original PAPI of 47.55%. On TINY IMAGENET, PICO$^+$$_{SAT}$ reaches an accuracy of 35.16% for $p=0.3$, a marked improvement over PICO$^+$'s 7.74%. These results suggest that integrating $\ell^{SAT}$ can significantly increase the robustness and performance of the existing frameworks under challenging high-label ambiguity scenarios.

**Results on Fine-grained Datasets.** Table 1 also shows that $\ell^{SAT}$ leads to consistently higher accuracy against state-of-the-art across varying levels of label ambiguity on fine-grained, complex datasets. Notably, PAPI$_{SAT}$ achieves exceptionally high accuracy on OXFORD FLOWER 102, reaching up to 95.34% for $q=0.1$, maintaining high accuracy as the partial label rate increases. Similarly, the integration of $\ell^{SAT}$ into PICO$^+$ significantly improves upon PICO$^+$ on CUB-200, where it records a notable accuracy of 51.28% for $q=0.15$. These results underline the high resilience of $\ell^{SAT}$ in challenging fine-grained contexts where the candidate labels come from subcategories (i.e., Raven, Crow, Blackbird) of the same class (e.g., bird).

**SAT loss helps learn more distinguishable representations.** In Figure 4, we illustrate the image representation encoded by the ResNet18 model using t-SNE (Van der Maaten & Hinton, 2008). We use the instance-dependent partial labeled CIFAR-10 data as the training set and present four approaches. Figure 4(a) and (b) shows the result from PICO$^+$ and its $\ell^{SAT}$-based counterparty PICO$^+$$_{SAT}$. Similarly, Figure 4(c) and (d) show the results of PAPI and PAPI$_{SAT}$, respectively. It

can be observed that the integration of $\ell^{SAT}$ greatly enhanced the representation quality, producing well-separated clusters. This is because the classifier and encoder share the same parameters in PICO$^+$ and PAPI, and integrating $\ell^{SAT}$ greatly enhanced the classifier performance. Therefore, we see the more distinguishable representations.

**Results for Instance-Dependent Partial Labels.** From Table 2, we can see that $\ell^{SAT}$ demonstrates superior performance, achieving the second highest accuracy of $89.32\%$ on CIFAR-10 and $56.93\%$ on CIFAR-100 without integrating with other techniques. The incorporation of $\ell^{SAT}$ into PICO$^+$, PAPI, and DPLL enhances their performance by a large margin. We can see that PAPI$_{SAT}$ and PICO$^+_{SAT}$ show remarkable improvements over their standalone counterparts. The consistent improvements across different baselines and benchmarks confirm the utility of $\ell^{SAT}$ in improving classification outcomes in different complex label scenarios, verifying the effectiveness of our satisfiability-based PLL reduction.

| METHODS | CIFAR-10 | CIFAR-100 |
|---|---|---|
| CC | 82.92% | 35.42% |
| RC | 83.61% | 44.56% |
| LWC | 88.04% | 46.99% |
| SAT | **89.32%** | **56.93%** |
| PAPI | 85.87% | 54.46% |
| PAPI$_{SAT}$ | **90.10%** | **57.66%** |
| PICO$^+$ | 83.54% | 56.62% |
| PICO$^+_{SAT}$ | **86.52%** | **57.74%** |
| DPLL | 71.46% | 35.46% |
| DPLL$_{SAT}$ | **74.74%** | **36.02%** |

Table 2: Mean classification accuracy for instance-dependent partial labels. **Best results** across all baselines are in red.

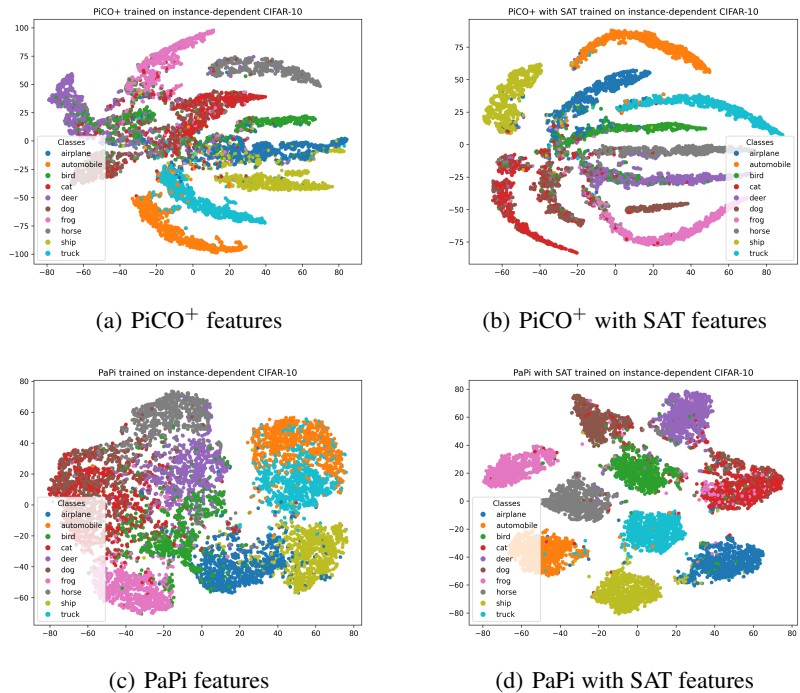

(a) PiCO$^+$ features

(b) PiCO$^+$ with SAT features

(c) PaPi features

(d) PaPi with SAT features

Figure 4: t-SNE visualization of the image representation on CIFAR-10(instance-dependent).

## 7 CONCLUSIONS

To conclude, this paper introduces a novel approach to Partial Label Learning (PLL) by framing it as a satisfiability problem and introducing a SATisfiability-based (SAT) loss function. Through both theoretical and empirical analyses, we demonstrate that this method offers substantial improvements in classification accuracy, especially in scenarios with high label ambiguity and fine-grained categories. Our SAT loss does not rely on pseudo labels and helps the model focus on low-entropy solutions, resulting in better label disambiguation. This work provides a solid foundation for future research on applying logical constraints to weakly supervised learning, with potential extensions to more complex and real-world datasets.

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
