# OpenReview forum: "Enhancing Deep Partial Label Learning via Casting it to a Satisfiability Problem"
_ICLR.cc/2025/Conference — ICLR 2025 Conference Withdrawn Submission_

### Official Review · Reviewer_Ej8p · 2024-10-16

**Soundness:** 1
**Presentation:** 2
**Contribution:** 1
**Rating:** 3
**Confidence:** 5

**Summary:**

This work discussed the partial label learning(PLL) problem, which means that the supervision information is a set of possible labels.

Specifically, this paper proposed a satisfiability-based(SAT) loss function to handle the PLL problem.

To do this, they first transform the partial label information into the propositional logic formula, e.g., $\displaystyle\phi = (\land_{i\in s} X_i)\land \neg(\text{more than one label is correct})$.
Then they optimize this loss function as any other loss function, by maximizing the semantic loss form.

The authors also established a generalization error bound under the low-ambiguity assumption.

**Strengths:**

+ The paper is generally easy to read and follow.
+ The proposed method shows a good performance.
+ They give a classical generalization error analysis.

**Weaknesses:**

- **Motivation is UNCLEAR.** The introduction failed to explain why this paper should introduce the SAT loss. As the authors claim that the previous works are not good enough, they say the previous methods have *undesired behavior*, which is very fun, because no method can guarantee error-free, especially in such a weakly-supervised learning scenario.
  - For example, the motivation example and analysis of the introduction part (see lines 27 to 100) can be used to introduce the motivation of any algorithm of partial label learning.
  - The necessity of transforming a partial label constraint to a propositional logic formula is unclear, why?

- **Claims are UNSUPPORTED or TRIVIAL.** There are two claims of the proposed method.
  - *Robustness*. The authors claim that the proposed SAT loss is robust to high label ambiguity. They explained that
> the SAT loss holds the property that its risk is non-increasing as the small ambiguity degree increases.

    This is not a good explanation, because *any* loss function with $\prod_{i \in s} \ell(f(x_i))$ form will satisfy this property.

  - *Disambiguation*. The authors say that the proposed SAT loss can perform better by *low entropy preference*. However, it is a small trick to regularize the model's output to have low entropy in the field of weakly-supervised learning. It is *not a rare property, many loss functions can induce such a property*, and it can not explain why this method is better since there is no more information used.

- **Theoretical analysis is EMPTY**. The given bound is not rigorous, first, the given bound can only hold when the classification task is binary, that is to say, this bound is not consistent with the notations introduced in the paper. Although this problem can be fixed easily to extend to a multiclass case. The established bound is no better than the earlier proposed PLL bound [1,2]. Especially, we can see that the Rademacher complexity is increased if we choose the SAT loss. Also, the established process of this error bound is very common in this field, thus can not bring any new insight or perspective to us.

- **Experiment results are Weried**. The reported performance of SAT loss is surprisingly high, for example, the reported performance of CIFAR-100 (see Table 1) is even better than the supervised case (purely use cross entropy). It is convinced that the experiments are done in unfair settings, i.e., using bags of tricks. Thus it is difficult to say the SAT loss is better. Besides, I *checked the code*, and the running code is different compared with what the authors said in their paper. For example, they use more than SAT loss, but also EMA trick, warmup learning rate decay, and mixup loss, and even more I have not carefully checked.  It is easy to do such a good performance if you can do many bags of tricks.


Refs:

---
[1] Timothee Cour, Ben Sapp, and Ben Taskar. 2011. Learning from Partial Labels. J. Mach. Learn. Res. vol.12, pp. 1501–1536.

[2] Lei Feng, Jiaqi Lv, Bo Han, Miao Xu, Gang Niu, Xin Geng, Bo An, and Masashi Sugiyama. 2020. Provably consistent partial-label learning.  In Advances in Neural Information Processing. Systems 33, pp. 10948–10960, Virtual, 2020.

**Questions:**

Q1: Can you explain your motivation clearly?

Q2: Can you support your claims?

Q3: Can you explain the superiority of your error bound compared with other papers (for example, the listed two)?

Q4: It is well-known that the semantic loss minimizes the inconsistency of the model and the logic constraint, the proposed loss can be seen as a special case of semantic loss. But the computational complexity of the semantic loss is very high, in fact, the computation of SL is #P, which is usually harder than NP. Can you compare the efficiency of the algorithm with other methods? Especially on a large scale (concerning the label space size).

---

> ### Author Response · Authors · 2024-11-19
> **Author Rebuttal**
>
> Thank you for your review comments. We are happy to address your questions and resolve some of the misunderstandings. We appreciate your time to read our responses.
>
> ### **Experiment**
>
> The accusation of experiment manipulation is a misunderstanding. For clarity, our best performance on CIFAR-100 was 81.16\%, whereas the supervised learning benchmark is 82.31\% as reported by [7].
> The so-called "using bags of tricks" was introduced in [5,6,7]. These training techniques are found in their open-source code. We adopted these techniques in our experiments to ensure a fair comparison with prior work.
>
> ### **Motivation**
>
> Discovering limitations in existing work, analyzing the reasons, and proposing new solutions is the fundamental logic of scientific research. On this point, we are in agreement with you. However, the challenges and analyses we identified in the *Introduction* section offer a unique perspective not previously proposed in earlier works. Specifically, we observed that state-of-the-art methods [5,6,7] experience severe performance drops in realistic scenarios involving high label ambiguity and fine-grained categorization. The core reason for this is that existing methods rely heavily on cross-entropy loss and pseudo-labels for model training. However, pseudo-labels generated via prototype learning are exceptionally noisy. We elaborated on this in lines 101-107, the paragraph starting with *"What is the root cause of this undesired behavior?"*.
>
> Our SAT loss, constructed through logical constraints, eliminates the dependence on pseudo-labels and does not require additional modules for training or any assumption on prior distribution. SAT loss integrates seamlessly into existing PLL frameworks and substantially mitigates the challenges caused by high label ambiguity and fine-grained categorization in experiments. We hope this clarifies our motivation. We will further improve the *Introduction* section to make it clear and comprehensible.
>
> ### **Claims**
>
> We believe the logic in the review comment here warrants further discussion. The success of SAT loss is due to the unique combination of multiple properties.
>
> **Robustness:**
>
> The existence of other loss functions conforming to the $\prod l(f(x))$ form does not disprove that $\prod l(f(x))$ supports Corollary 4.2. Additionally, previous loss function designs in the PLL domain have predominantly adopted forms similar to $\sum l(f(x))$, which further underscores the uniqueness and importance of SAT loss as a robust loss according to Corollary 4.2.  This is noted in the second paragraph of Sec. 4.1.
>
> **Disambiguation:**
>
> Once again, the existence of other loss functions sharing similar characteristics does not render the *low-entropy preference* property useless. The following example effectively illustrates why SAT loss benefits label disambiguation.
>
> In a multi-class classification problem with classes A, B, C, and D, consider the data instance $x$ with the partial label $y = \\{0, 1, 0, 0\\}$, where B is the ground-truth label. For two predicted probability distributions:
> $$
> f_1(x) = \\{p_1^A = 0.25, p_1^B = 0.25, p_1^C = 0.25, p_1^D = 0.25\\}
> $$
> $$
> f_2(x) =\\{p_2^A = 0.01, p_2^B = 0.97, p_2^C = 0.01, p_2^D = 0.01\\},
> $$
> we calculate their corresponding SAT loss values as:
> $$
> l_{\text{SAT}}(f_1(x)) = 0.38, \quad l_{\text{SAT}}(f_2(x)) = 0.0295.
> $$
> This demonstrates that SAT loss encourages the model to concentrate predicted probabilities on a single label, thereby promoting label disambiguation. In contrast, pseudo-labels from prior works introduce noise into model training due to high label ambiguity and fine-grained categorization.
>
> This is also different from regularizing other losses with low-entropy regularizers, where one would need to introduce yet another regularization hyperparameter to impose low-entropy; instead, SAT loss does this intrinsically.

---

> ### Author Response · Authors · 2024-11-19
> **continue**
>
> ### **Theoretical Analysis**
>
> **Binary Classification**
>
> The error bound proposed in Section 5.3 is consistent with our notation. We did not assume binary classification conditions.
>
>
> **Compared to other methods**
>
> The error bound shown in our paper is the generalization bound, which indicates the SAT loss has good learnability and generalization ability. What you intend to talk about (the so-called "PLL bound") is the convex upper bound, which is an entirely different thing.
>
> For the two previous papers you mentioned [8, 12]:
>
> [8] is an early solid work on PLL. It provides a convex upper bound comparison in proposition 6 as there are very few PLL works to compare at that time. We are happy to offer you the proof showing that CLPL [8] is an upper bound for SAT loss, i.e. SAT loss is better than CLPL. See the full proof at the end of this rebuttal reply.
>
>
> [12] is one of our baseline methods, i.e., RC and CC. We have provided extensive experiments to demonstrate the superiority of SAT loss over them. This provides empirical evidence that our method is better under the same learning context. [12] is essentially similar to [2], both based on Bayesian Inference. A similar proof can be made based on the proposition in [8] and the proof we provided. Additionally, [4] did not explicitly include the impact of a small ambiguity degree in their error bound, which we did.
>
> **Rademacher complexity**
>
> To further address your comment regarding the complexity of hypothesis space, we will add a new perspective to our analysis that stresses the pros of the
> SAT loss. Recall that when a loss function is minimized, the goal is to find the optimal classifier within the
> search space that best fits the training data. In the following we will show the optimization of SAT loss will lead to fewer classifiers, compared to the NLL loss[4] and cross-entropy loss[6,7]. This is advantageous because it simplifies the learning process and reduces the likelihood of overfitting. In other words, **SAT loss leads to a smaller Rademacher complexity.**
>
> The NLL loss ensures that if the small ambiguity degree holds, then the classifier that minimizes the NLL loss
> over the training data distribution ($\mathcal{X} \times \mathcal{S}$, where $\mathcal{S}\subseteq 2^{\mathcal{Y}}$), will map each instance to its true label, which applies to SAT loss as well.
>
>
> However, in the real world, the confusing candidate labels can co-occur with the true label. All the Raven images
> may share the same candidate label set {"Raven", "Crow", "Blackbird", "Grackle"}. When the small ambiguity
> degree is violated, the NLL loss leads to a larger space of (erroneous) classifiers than the SAT loss. As
> shown in this four-class case $\\{1,2,3,4\\}$. We have two training samples $(x, \\{1,2,3\\}),(x, \\{1,2,4\\})$. The true
> label for $x$ is $1$. Given three different classifiers:
>
> - $f(x) = \\{0.5,0.5,0,0\\}$(maps to class 1 with 0.5 probability and to class 2 with 0.5 probability)
>
> - $f'(x) = \\{1,0,0,0\\}$(maps to class 1 with 1 probability)
>
> - $f''(x) = \\{0,1,0,0\\}$(maps to class 2 with 1 probability)
>
> In contrast, classifier $f$ cannot minimize the expected error over the training data under that loss, resulting in a space with fewer erroneous classifiers. The cross-entropy loss leads to fewer classifiers than the NLL loss. However, as the gold label is unknown to the learner, [5,6,7] employ the cross-entropy loss with pseudo-labels that are computed using the classifiers' predictions. Doing so, these pseudo-labels may greatly deteriorate the training of the classifier as our empirical analysis shows and was demonstrated in Figure 2. For [2], their objective is to maximize the probability that the predicted label is within the ground-truth class. According to Eq.(1) in [2], that probability is computed by taking the product of the classifier’s predictions. This formulation also leads
> to many erroneous classifiers. In terms of the example we presented above about the NLL loss, the formulation
> of Eq. (1) in [2] will also lead to all those classifiers $f$, $f'$, and $f''$. This is because the logarithm is taken outside of the
> product. For the same reason, Eq. (1) in [2] will favor uniform distributions, in contrast to the SAT loss and the cross-entropy one which favors concentrated distributions. Closing our reply, it is straightforward to see that the SAT loss reduces to the general ambiguous loss [3, 8] in the PLL literature. Thus, the SAT loss reduces Rademacher complexity in the real world.
>
> We will include the above discussion in a revised version of our work.

---

> > ### Author Response · Authors · 2024-11-19
> > **continue**
> >
> > **Established Process**
> >
> > The established process of deriving error bounds is written in machine learning textbooks such as [14]. Establishing the error bound aims to demonstrate the learnability and generalization ability of a loss function. Using new mathematical tricks to reinvent the proof process is not our contribution or focus in this work. It is also worth mentioning that the paper [12] mentioned by you claimed they used the same proofing/establishing process as [14] see their appendix for details.
> >
> > ### **Computational Complexity**
> >
> >
> > It is true for semantic loss the worst case could be up to \#P-hard. However, for SAT loss, the primary computational cost arises from the product operation over the set of candidate labels $s$. **This operation is linear concerning the size of $s$, denoted as $|s|$.** The overall complexity per sample is $\mathcal{O}(|s|)$ for computing the product. Given that this needs to be done for every data point in a dataset of size $N$, the computational complexity becomes $\mathcal{O}(N|s|)$. Similarly, we can get the computational complexity for cross-entropy loss as $\mathcal{O}(NC)$, where $N$ is the number of the data samples and $C$ is the number of the classes ($C > |s|$).
> >
> > For a further discussion, as shown in [16], if the input propositional formula is both iDNF and 1OF (each variable occurs exactly once), then the computation of its probability (that is weighted model counting in our case) can be done by simply scanning the formula. Indeed as stated in Section 2 of that paper “In contrast, formulas in 1OF and iDNF admit probability computation in linear time.” (this text can be found in the paragraph under formula 3). Going back to our case, the propositional formula that expresses that either of the labels is the correct one is both iDNF and 1OF (excluding the mutual exclusiveness constraint).
> >
> > In practice, integrating SAT loss did not bring any noticeable extra training time. Therefore, the computational complexity of SAT loss is not a big concern.

---

> ### Author Response · Authors · 2024-11-19
> **Proof on the convex upper bound relationship between SAT loss and CLPL[8]**
>
> We aim to prove that CLPL [1], i.e. $\ell_\psi^{max}$, provides an upper bound for $\ell_{SAT}$. For a partial label instance $x$, its candidate label set is $s$. The model $f()$ gives its prediction on class $j$ by $f_j(x)$.
>
> Given the definition of SAT loss:
> $$\ell_{SAT}(f(x),s) = -log(1-\prod_{j\in s}(1-f_j(x))$$
>
> Definition of CLPL loss:
> $$
> \ell_\psi^{\text{max}}(f(x), \mathbf{s}) = \psi\left(\max_{j \in s} f_j(x)\right) + \sum_{j \notin s} \psi\left(-f_j(x)\right).
> $$
>
> where for log loss, $\psi(z) = log(1+e^{-z})$.
>
> Our goal is to prove:
>
> $$
> \ell_{\text{SAT}} \leq \ell_\psi^{\text{max}}(f(x), \mathbf{s}).
> $$
>
>
> We first bound the product term in $\ell_{SAT}$.
> The term $\prod_{j \in s} (1 - f_j(x))$ represents the probability that **all** correct labels are **not** predicted. Since $0 \leq f_j(x) \leq 1$, we know that  $0 \leq 1 - f_j(x) \leq 1 $.
>
> It can be observed that the product of numbers in $ [0, 1] $ is less than or equal to the smallest number:
>
> $$
> \prod_{j \in s} (1 - f_j(x)) \leq \min_{j \in s} (1 - f_j(x)).
> $$
>
> Thus, we have:
>
> $$
> 1 - \prod_{j \in s} (1 - f_j(x)) \geq 1 - \min_{j \in s} (1 - f_j(x)) = \max_{j \in s} f_j(x).
> $$
>
>
> Since the logarithm is monotonically increasing, applying $ -\log $ to both sides:
>
> $$
> -\log\left(1 - \prod_{j \in s} (1 - f_j(x))\right) \leq -\log\left(\max_{j \in s} f_j(x)\right).
> $$
>
> Therefore, we have **Equation 1** as:
>
> $$
> \ell_{\text{SAT}} \leq -\log\left(\max_{j \in s} f_j(x)\right).
> $$
>
>
> Further, we know that the log loss function $ \psi(z) = \log(1 + e^{-z}) $ satisfies the following inequality for $ z > 0 $:
>
> $$
> -\log(z) \leq \log(1 + e^{-z}) = \psi(z).
> $$
>
> Thus, we can further bound $ \ell_{\text{SAT}} $ as:
>
> $$
> \ell_{\text{SAT}} \leq \psi\left(\max_{j \in s} f_j(x)\right).
> $$
> This is noted as **Equation 2**.
>
>
> Now let's look at CLPL.
> The $ \ell_\psi^{\text{max}} $ loss includes a summation over incorrect labels:
>
> $$
> \ell_\psi^{\text{max}}(f(x), \mathbf{s}) = \psi\left(\max_{j \in s} f_j(x)\right) + \sum_{j \notin s} \psi\left(-f_j(x)\right).
> $$
>
> Since $ \sum_{j \notin s} \psi\left(-f_j(x)\right) \geq 0 $, we know:
>
> $$
> \ell_\psi^{\text{max}}(f(x), \mathbf{s}) \geq \psi\left(\max_{j \in s} f_j(x)\right).
> $$
>
> This is noted as **Equation 3**.
>
>
> From **Equations 2 and 3**, we conclude:
>
> $$
> \ell_{\text{SAT}} \leq \psi\left(\max_{j \in s} f_j(x)\right) \leq \ell_\psi^{\text{max}}(f(x), \mathbf{s}).
> $$
>
> Thus, we have shown that:
>
> $$
> \ell_{\text{SAT}}(f(x), \mathbf{s}) \leq \ell_\psi^{\text{max}}(f(x), \mathbf{s}),
> $$
>
> which demonstrates that $ \ell_\psi^{\text{max}} $ provides a convex upper bound for $ \ell_{\text{SAT}} $.

---

> ### Comment · Reviewer_Ej8p · 2024-11-20
>
> Thanks for the detailed response of the authors, after reading the response, some of my concerns are addressed, but I still have a few questions. It would be great if you could help me address the following questions.
>
> **1. Motivation and Use of Pseudo-Labels**
>
> Thank you for the detailed response and code sharing. After reviewing both the response and the code, I still have some lingering concerns regarding the use of pseudo-labels in the method.
>
> From my understanding, the motivation behind this work is that previous methods heavily rely on pseudo-labels and cross-entropy loss, which can result in sub-optimal performance, especially in high ambiguity cases. However, after reviewing the implementation, I noticed that the code still uses pseudo-labels (as shown in `Model.py`), even though the authors mention that pseudo-labels are noisy and may cause performance issues. Specifically, the authors mention:
>
> > "Pseudo-labels generated via prototype learning are highly noisy... SAT loss eliminates the dependence on pseudo-labels."
>
> This raises a question: Why does the method still use pseudo-labels in the forward pass if they can introduce noise into the training process? Could the authors clarify this apparent contradiction?
>
> **2. SAT Loss and Its Advantage**
>
> While I understand that the SAT loss has two key properties, *disambiguation* and *robustness*, I still struggle to see why this makes the method fundamentally better than other loss functions. As pointed out in my review, the term *robustness* can often be easily constructed, and I believe the authors should elaborate on the specific advantages SAT loss brings.
>
> Moreover, the claim that SAT loss is better than existing methods is somewhat unsubstantiated from a theoretical perspective. As I pointed out, simply showing that one loss function $\ell_a$ is less than or equal to another loss function $\ell_b$ doesn't necessarily imply that $\ell_a$ is superior. An example would be $\ell_b := 0.9\ell_a$, which would trivially hold but does not reflect a meaningful improvement.
>
> Could the authors provide a higher-level explanation of why SAT loss offers superior performance over other methods, particularly addressing its *disambiguation* and *robustness* properties more clearly?
>
> **3. Rademacher Complexity**
>
> Lastly, I remain unclear about the authors' claim that SAT loss has a lower Rademacher complexity than other methods, as no direct comparison is provided in the response. To clarify this point, I believe a mathematical explanation would be very helpful.
>
> Could the authors provide the mathematical formulation of Rademacher complexity and compare it for SAT loss against other methods? A direct mathematical derivation or a detailed explanation of why SAT loss is said to lower this complexity would greatly help in understanding the reasoning behind this claim.
>
>
> I am very appreciate the efforts the authors make, and I will be glad to increase the score if the above key concerns are addressed properly.

---

> ### Author Response · Authors · 2024-11-21
> **Following Rebuttal**
>
> Thanks for your prompt response and your time in reading our rebuttal. We are pleased to see these constructive discussions gradually help us build consensus. Here is our response to the remaining questions.
>
> ### **Pseudo Label in Code**
>
> In the `forward` pass in the `model.py`, two lines of code are computing the pseudo label. However, with one more look, it can be found that the generated `pseudo_labels_b` will not be used anywhere after. It will not be involved in further computation or returned. **Therefore, we are not using pseudo labels in our method.**
>
> It was there because we conducted some analysis in the past. We will delete those two lines of code in the open-source version for better clarity.
>
> ```
> def forward(self, img_q, im_k=None, Y_ori=None, args=None, eval_only=False, stop_warmup=False):
>
>         output, q = self.encoder_q(img_q)
>         if eval_only:
>             return output
>
>         with torch.no_grad():  # no gradient
>
>             predicetd_scores_q = torch.softmax(output, dim=1)
>             within_max_cls_conf, within_max_cls = torch.max(predicetd_scores_q * Y_ori, dim=1)
>             all_max_cls_conf, all_max_cls = torch.max(predicetd_scores_q, dim=1)
>
>             pseudo_labels_b = within_max_cls
>             pseudo_labels_b = pseudo_labels_b.long()
>
>             predicetd_scores_q = predicetd_scores_q * Y_ori
>             predicetd_scores_q = predicetd_scores_q / predicetd_scores_q.sum(dim = 1).repeat(args.num_class, 1).transpose(0, 1)
>             predicetd_scores_k = torch.softmax(output, dim=1) * Y_ori
>             predicetd_scores_k = predicetd_scores_k / predicetd_scores_k.sum(dim = 1).repeat(args.num_class, 1).transpose(0, 1)
>             predicetd_scores = torch.cat((predicetd_scores_q, predicetd_scores_k),dim=0)
>
>         return output, predicetd_scores,  all_max_cls_conf
> ```

---

> ### Author Response · Authors · 2024-11-21
> **continue**
>
> ### **Advantages of SAT loss**
>
> We would like to discuss:
>
> 1. What are the Robustness and Disambiguation properties of SAT loss?
>
> 2. How can we demonstrate SAT loss is better than the predecessors combined with these two properties?
>
> **Disambiguation (Low Entropy Preference)**
>
> Label disambiguation essentially means the process of finding the true label from the candidate label set. In standard datasets like CIFAR-10 or MNIST, the model could easily distinguish different candidate labels, such as \\{dog, cat\\}, which may result in high model confidence $\\{f_{dog}=0.9, f_{cat}=0.05,...\\}$. The max model confidence is as high as $0.9$ in this case, indicating the data sample is well disambiguate. Previous methods[5,6,7] try to generate pseudo labels in such a case is acceptable as the learning task is easy.
>
>
> However, for fine-grained categorization, the task is more challenging. The model could struggle to distinguish Raven and Crow, resulting in tied prediction $\\{f_{Raven}=0.45, f_{Crow}=0.45,...\\}$ or wrong predictions $\\{f_{Crow}=0.55, f_{Raven}=0.4,...\\}$. This will lead to noisy pseudo labels and performance drops for previous state-of-art methods. In other words, the data could not be well disambiguated due to the difficulty brought by fine-grained labels.
>
> The SAT loss solves this problem by its *Low Entropy Preference* property. We provide the complete proof in the Appendix. while to better understand, let's look at the toy example we provided in the previous rebuttal.
>
> The training process can be seen as a search for a better classifier in the hypothesis space based on a loss function. In a multi-class classification problem with classes A, B, C, and D, consider the data instance $x$ with the partial label $y = \\{0, 1, 0, 0\\}$, where B is the ground-truth label. For two predicted probability distributions:
> $$
> f^1(x) = \\{p_1^A = 0.25, p_1^B = 0.25, p_1^C = 0.25, p_1^D = 0.25\\}
> $$
> $$
> f^2(x) =\\{p_2^A = 0.01, p_2^B = 0.97, p_2^C = 0.01, p_2^D = 0.01\\},
> $$
> we calculate their corresponding SAT loss values as:
> $$
> l_{\text{SAT}}(f^1(x)) = 0.38, \quad l_{\text{SAT}}(f^2(x)) = 0.0295.
> $$
>
> We can see SAT loss can correctly choose the classifier which can well disambiguate the data. While the loss function $\ell_1 = -log(\prod_{j\in s}f_j(x))$ and $\ell_2 = -log(\prod_{j\notin s}(1-f_j(x)))$ will choose the opposite. The *Low Entropy Preference* demonstrates that SAT loss encourages the model to concentrate predicted probabilities on a single label, thereby promoting label disambiguation.
>
> This is an advantage of SAT loss and can be verified:
>
> 1. In Figure 2(f) of our paper, integrating SAT loss leads to higher average maximum model confidence, indicating better disambiguation.
>
> 2. The theory justification from the Rademacher complexity analysis provided at the later of this rebuttal reply. See **Impact of Model Confidence**, we show that as the model confidence increases, the SAT loss maintains a smaller Rademacher complexity, i.e. fewer classifiers. This is advantageous because it simplifies the learning process and reduces the likelihood of overfitting.
>
> 3. Our superior experiment results.
>
>
> **Robustness (Monotonicity under ambiguity)**
>
> The robustness of SAT loss here indicates it brings robustness to the model when the label ambiguity is high. Essentially, a high label ambiguity means the size of the candidate label set $s$ is large. The advantages of SAT loss relating to this property can be shown from three aspects:
>
> 1. The loss value of SAT loss is robust to the high label ambiguity.
>
> Based on Corollary 5.2, we show that the internal formula structure $1-\prod_{j\in s}(1-f_j(x))$ brings this property. A loss function in the form of $\prod \ell(f(x),s)$ can not guarantee this as the value of $\ell(f(x),s)$ does not always belong to $(0,1)$.  Other loss functions like $\ell_1 = -log(\prod_{j\in s}f_j(x))$ and $\ell_2 = -log(\prod_{j\notin s}(1-f_j(x)))$ has the similar property but does not have *Low Entropy Preference*. In addition, the representative PLL work in the past decades [2,3,4,5,6,7,8,9,10,11,12] did not propose a loss with either of the properties mentioned above.
>
> As we discussed, the unique combination of all these properties guarantees the effectiveness of SAT loss. It may be not difficult to find loss functions that share one of the properties but construct a loss function to have both the *Monotonicity under ambiguity* and *Low Entropy Preference*.
>
>
> 2. The SAT loss maintains a robust and smaller Rademacher complexity compared to loss as Cross-Entropy Loss.
>
> As shown in **Impact of Ambiguity Degree**, we analyze how the increase of candidate label set size will impact the Rademacher complexity of SAT loss and Cross-Entropy Loss, including the extreme cases when the $|s| \to \infty$ and $|s| = 1$. The SAT loss always has a smaller Rademacher complexity when $|s| \geq 2$ (general PLL). And we have discussed why smaller Rademacher complexity is good.

---

> > ### Author Response · Authors · 2024-11-21
> > **continue**
> >
> > 3. Again, our experiment results show that SAT loss significantly improve the model's performance under high ambiguity degree scenarios by up to $25$\%.
> >
> >
> > >Moreover, the claim that SAT loss is better than existing methods is somewhat unsubstantiated from a theoretical perspective. As I pointed out, simply showing that one loss function $\ell_a$ is less than or equal to another loss function $\ell_b$ doesn't necessarily imply that  is superior. An example would be $\ell_b := 0.9\ell_a$, which would trivially hold but does not reflect a meaningful improvement.
> >
> > We also would like to answer to this point. We agree that simply showing that given the same input, the values of $\ell_a$ are no greater than $\ell_b$ can not prove $\ell_a$ is better than $\ell_b$. And we assume you want to say $\ell_b := 0.9\ell_b$.
> >
> > The proof we showed in the previous rebuttal is proof to show the CLPL loss is a convex upper bound of SAT loss. It should combined with Proposition 6 in [8] and  Lemma A.5 to show that SAT loss is a tighter surrogate loss of partial 0/1 loss compared to CLPL loss[8]. This indeed shows SAT loss is one step closer to the optimal minimum. However, we apologize for not explaining this clearly.

---

> ### Author Response · Authors · 2024-11-21
> **continue**
>
> ### **Rademacher Complexity**
>
>
> **Proof on the Rademacher Complexity Bound Analysis of SAT Loss and Cross-Entropy Loss**
>
> In a partial label learning problem with totally $C$ classes, for a data instance $x$, its candidate label set is $s$, with a size $|s|$. We define $\ell(f(x),s)$ as a loss function, where $f$ is a classifier belonging to a hypothesis class $\mathcal{H}$. We define the Lipschitz constant [20] of loss function $\ell$ as $L$.
>
> Given a sample size $n$, the Rademacher complexity of a loss function $\ell$ applied to a hypothesis class $\mathcal{H}$ is defined as:
>
> $$
> \Re_{n}(\ell \circ \mathcal{H}) =\mathbb{E}_{\sigma}\left[ \sup_{f \in \mathcal{H}} \frac{1}{n} \sum_{i=1}^n \sigma_i \ell(f(x_i), s_i) \right]
> $$
>
> where $\sigma_i \in \{0,1\}$ are the Rademacher random variables.
>
>
> Next, we analyze the Rademacher complexities of SAT loss and Cross-Entropy Loss.
>
> ***
>
> **Lipschitz Constant of SAT loss**
>
> Given the definition of SAT loss as:
>
> $$
> \ell_{SAT}(f(x),s) = -log(1-\prod_{j\in s}(1-f_j(x))
> $$
>
> where $f_j(x)\in [0,1]$ is the predicted probability for class $j$ and $s$ is the candidate label.
>
> Let's first compute the gradient of $\ell_{SAT}$ with respect to $f_j(x)$. We have:
>
> $$
> \frac{\partial L_{SAT}(f(x), s)}{\partial f_j(x)} = \frac{\prod_{k \in s \setminus \{j\}} (1 - f_k(x))}{1 - \prod_{k \in s} (1 - f_k(x))}.
> $$
>
> where the numerator $\prod_{k \in s \setminus \{j\}} (1 - f_k(x))$ represents the product of $1 - f_k(x)$ for all $k \neq j$. The denominator $1 - \prod_{k \in s} (1 - f_k(x))$ is the argument of the logarithm.
>
>
> The gradient's magnitude is maximized when:
>
> 1. $f_k(x) \to 0$ for all $k \in s \setminus \{j\}$, making the numerator close to $1$.
>
> 2. $1 - \prod_{k \in s} (1 - f_k(x))$ is close to $0$, which occurs when $f_k(x) \to 0$ for all $k \in s$.
>
>
> Thus, the Lipschitz constant is:
> $$
> L_{SAT} = \sup_{f(x)} \frac{1}{1 - \prod_{k \in s} (1 - f_k(x))}
> $$
>
> In practice, $f_k(x) \in [\epsilon, 1]$ for some small $\epsilon > 0$, so:
> $$
> L_{SAT} = \frac{1}{1 - (1 - \epsilon)^{|s|}}
> $$
>
> where $|s|$ is the size of the candidate label set.
>
> ***
>
> **Lipschitz Constant of Cross-Entropy Loss**
>
> Given the definition of Cross-Entropy Loss, we have:
> $$
> \ell_{CE}(f(x),s) = -\sum_{j\in s}log(f_j(x))
> $$
> where $f_j(x)\in[0,1]$.
>
> Differentiating with respect to $f_j(x)$:
>
> $$
> \frac{\partial L_{CE}(f(x), s)}{\partial f_j(x)} = -\sum_{j\in s}\frac{1}{f_j(x)}
> $$
>
> The gradient magnitude is maximized when $f_j(x)$ is minimized. For realistic case, when $f_j(x) \in [\epsilon,1]$, the Lipschitz constant is:
>
> $$
> L_{CE} = \sum_{j \in s} \frac{1}{\epsilon} = \frac{|s|}{\epsilon}.
> $$
>
>
> ***
>
> Based on Kakade \& Tewari Lemma [21], we have
>
> $$
> \Re_n(\ell \circ \mathcal{H}) \leq L \Re_n(\mathcal{H})
> $$
>
> Therefore, we have
>
> $$
> \Re_n(\ell_{SAT} \circ \mathcal{H}) \leq L_{SAT} \Re_n(\mathcal{H})
> $$
>
> $$
> \Re_n(\ell_{CE} \circ \mathcal{H}) \leq L_{CE} \Re_n(\mathcal{H})
> $$
>
> Based on the definition of Rademacher complexity, we are looking for the expectation of the supremum (the least upper bound) of all functions $f\in \mathcal{H}$ over the Rademacher distribution. The supremum involved in Rademacher complexity scales linearly with the Lipschitz constant of the loss function. This means that a loss function $\ell$ with a higher Lipschitz constant "stretches" the hypothesis space more, leading to a higher Rademacher complexity proportional to $L$. Therefore, we have:
>
> $$
> \frac{\Re_n(\ell_{SAT})}{\Re_n(\ell_{CE})} \propto \frac{L_{SAT}}{L_{CE}}
> $$
>
> Now, we build the relationship between the Rademacher complexity and the Lipschitz constant. We will discuss the relationship between $L_{SAT}$ and $L_{CE}$.
>
> **Impact of Ambiguity Degree**
>
> Ambiguity degree is directly associated with the candidate label set size $|s|$. A high ambiguity degree indicates a large candidate label size.
>
> - Case 1: In a large-scale dataset with nearly infinite classes and an ambiguity degree approaching to $1$. We have $|s|\to \infty$, and $\epsilon$ as a small number,then:
> $$
> \lim_{|s|\to \infty} L_{SAT} = \lim_{|s|\to \infty} \frac{1}{1 - (1 - \epsilon)^{|s|}} = 1
> $$
>
> $$
> \lim_{|s|\to \infty} L_{CE} = \lim_{|s|\to \infty} \frac{|s|}{\epsilon} = \infty
> $$
>
> We have $L_{CE} > L_{SAT}$, therefore $\Re_n(\ell_{CE}) > \Re_n(\ell_{SAT})$.
>
> This will hold for all $s$ with $|s| \geq 2$ as $\epsilon$ is a small number.
>
>
> - Case 2: Another extreme case is when $|s| = 1$, i.e. supervised learning. We have:
>
> $$
> L_{SAT} = L_{CE} = \frac{1}{\epsilon}
> $$
>
> Then, $\Re_n(\ell_{CE}) = \Re_n(\ell_{SAT})$.
>
> This is not a coincidence as the SAT loss turns to cross-entropy loss in a supervised learning setting.
>
>
> $$
> \ell_{SAT}(f_j(x),s) = -log(1-(1-f_j(x)))  = -log(f_j(x)) = \ell_{CE}(f_j(x),s)
> $$

---

> ### Author Response · Authors · 2024-11-21
> **proof continue**
>
> **Impact of Model Confidence $\epsilon$**
>
> During the training, the model will be more and more confident in its prediction, resulting in a larger $f_j(x)$. We discuss the case when $\epsilon \to 1$. We first define $\delta = (1-\epsilon)^{|s|}$.
>
> As $\epsilon \to 1$,
>
> $$
> \lim_{\epsilon \to 1} \delta = 0
> $$
>
> $$
> \lim_{\epsilon \to 1} L_{SAT} = \lim_{\epsilon \to 1} \frac{1}{1 - \delta} = 1
> $$
>
> $$
> \lim_{\epsilon \to 1} L_{CE} = \lim_{\epsilon \to 1} \frac{|s|}{\epsilon} \geq 1
> $$
>
> Still if and only if $|s| = 1$ (supervised learning), $\lim_{\epsilon \to 1} L_{CE} = 1$. For general partial label learning with $|s| \geq 2$, we have $\Re_n(\ell_{CE}) > \Re_n(\ell_{SAT})$.
>
> **End of the Proof**
>
> To close the proof, we have shown that For general partial label learning with $|s| \geq 2$, we have $\Re_n(\ell_{CE}) > \Re_n(\ell_{SAT})$. Thus, SAT loss always leads to a smaller Rademacher Complexity and has a better generalization to unseen data.

---

> ### Author Response · Authors · 2024-11-21
> **continue**
>
> ### **Revisit**
>
> In the discussion above, we have thoroughly elaborated on the strengths of our work in terms of motivation, methodology, theoretical analysis, and experiments. We are also delighted to see that the discussion with you has been highly constructive. Many of the questions you raised have provided valuable perspectives. We have addressed the advantages of SAT loss compared to prior works from various angles, such as Generalization Bounds, Convex Upper Bounds, Computational Complexity, and Rademacher Complexity, showcasing the technical soundness of our paper.
>
> Our work serves as a bridge between the fields of logical constraints and neural methods. The SAT loss constructed through logical relationships remarkably enables the transfer of logical constraint relationships into neural network training. To the best of our knowledge, this is the first of its kind in the PLL and even the weakly supervised learning community. We believe this work has the potential to inspire future research and extend to other fields such as multiple-label learning and complementary-label learning, demonstrating its broad impact.
>
> We hope that our response addresses your concerns and provides evidence that this is a good paper and a valuable contribution to the ICLR community. We sincerely appreciate your feedback and the time you have taken to review our work.

---

> > ### Comment · Reviewer_Ej8p · 2024-11-21
> >
> > Many thanks for the reply.
> > However, I still have a few concerns.
> >
> >  1. **Comparison Based on the Upper Bound:**
> >
> > I appreciate the explanation provided, but I believe there may be some confusion regarding the use of Rademacher complexity in this context. The comparison appears to focus on the Lipschitz constant rather than directly on the Rademacher complexity definition. This type of comparison is only valid when the upper bound is tight. For example, it is not accurate to conclude that "0.1 < 0.5 implies 0.7 < 0.6" based on loose upper bounds, such as in the case of "0.7 <= 0.1 * 1000" and "0.6 <= 0.5 * 1000". If the upper bound were tighter, the explanation would hold, but it is important to clarify this distinction.
> >
> > Moreover, the same insight applies to the surrogate loss comparison. A loss function being smaller (tighter) does not necessarily indicate it is better for the learning task. It is crucial to analyze whether the loss function contributes to smoother training, reduces variance, and lowers computational cost. These aspects might be more informative in evaluating the efficacy of a loss function. I suggest the authors expand on these considerations to provide a more thorough analysis.
> >
> > Additionally, the semantic loss function may face challenges in large label spaces. For instance, if the label space exceeds 100, values such as $0.9^{100}$ approach zero, which poses a significant issue during training. This is particularly critical in real-world scenarios and should be addressed. Based on my experience in the field, I recommend the authors provide more discussion on this limitation.
> >
> > ---
> >
> >  2. **Pseudo-label Issue:**
> >
> > While the authors claim that pseudo-labeling is not utilized in their approach, there is evidence in the `utils/util_loss.py` file indicating that the method employs pseudo-labels (or "confidence", as named in the code) during the warmup phase (before 100 epochs). Furthermore, the use of both CE loss and pseudo-labeling continues throughout the training process, especially with the Mixup loss. This appears to contradict the authors' assertion.
> >
> > Could the authors clarify whether this discrepancy is intentional, and explain the usage of pseudo-labels in the code? A more detailed analysis would be valuable to resolve this apparent inconsistency.
> >
> > ---
> >
> >  3. **Overclaim:**
> >
> > There is substantial research on the link between logical reasoning and machine learning, especially in areas like neuro-symbolic and abductive learning. Furthermore, neuro-symbolic learning can be seen as a type of weakly supervised learning.
> > However, this work seems unrelated to logical reasoning, so the authors should not claim credit in that context. The design of loss functions based on logical constraints, such as Semantic Loss from ICML 2018, is not new.

---

> > > ### Author Response · Authors · 2024-11-22
> > > **Following Rebuttal 2**
> > >
> > > Thanks for your reply. We are glad to see the previous concerns have been well addressed.
> > >
> > > ### **Q1:**
> > >
> > > We have explicitly built and discussed the relationship between the Rademacher complexity and the Lipschitz constant. If we can check back to the proof, "this distinction" has been presented in plain and straightforward language.
> > >
> > > Regarding the proof of surrogate loss, we have  provided an additional explanation in the previous reply on why our proof is valid to demonstrate that SAT loss is better than CLPL[8]. Specifically, Proposition 6 in [8] built a convex optimization analysis based on surrogate loss tightness to show that their proposed CLPL[8] is better than EM model[2], as CLPL is a tighter approximation of partial 0/1 loss. Our proof shows this analysis can be extended to include the SAT loss by building the relationship between SAT loss and CLPL loss and SAT loss with partial 0/1 loss (Lemma A.5 in our paper). A detailed discussion can be found in Section 4.3 of [8], which can be a good example to illustrate the loss comparison using a convex upper bound providing an approximation to the non-convex problem.
> > >
> > > Given $\ell_a$ and $\ell_b$, to prove that $\ell_b$ is a convex upper bound of $\ell_a$, we need to show that:
> > >
> > > 1. $\ell_b(x) \geq \ell_a$ (x), for all $x$
> > >
> > > 2. $\ell_b$ is convex. (note that CLPL[8] is convex.)
> > >
> > > The convex upper bound analysis between different loss functions and the different scaling of the same loss function is entirely different.
> > >
> > > - For comparison between loss functions, the scaling is meaningless. If $\ell_1$ times itself by $0.0001$, then $\ell_2$ can do that as well. The comparison should focus on the nature of the loss function.
> > >
> > > - For the comparison between the different scaling versions of the same loss function (assume it is convex), yes,  $\ell_a \geq 0.9\ell_a$ always holds and it satisfies the definition of convex upper bound. But it also does not change the characteristic brought by convex upper bound. Why? Because it is essentially turned into a training dynamic analysis. The nature of different scaling for the same loss function can be seen as choosing different learning rates. In a convex optimization landscape, a large learning rate may cause the model to repeatedly jump and oscillate around the global optimal point, failing to achieve the final optimization. While a small learning rate can ultimately lead you to the global optimal point but may cause slow convergence issues and require impractically large training iterations.
> > >
> > > For the above statement, a better justification can be found in [14,19,20].
> > >
> > >
> > > Lastly, the underflow issue. This issue applies to massive loss functions, including Cross-Entropy Loss, Kullback-Leibler Divergence, Exponential Loss, Hinge Loss, NLL Loss, etc. Specifically, for the underflow caused by $\prod$, there are a lot of techniques that can avoid or mitigate it in practice. A representative example could be the Advantage Normalization, which is commonly used in reinforcement learning (RL) to stabilize and improve the learning process. As in RL, the discounted reward usually contains a discounted factor $\gamma^t$, where $\gamma\in (0,1)$.

---

> > > > ### Author Response · Authors · 2024-11-22
> > > >
> > > > ### **Q2:**
> > > >
> > > > Once again, it is super clear that the SAT loss does not require pseudo labels. It can be seen from its math formula and the code implementation of this loss.
> > > >
> > > > The code base we provided is the framework to integrate SAT loss into baseline methods, such as PiCO+[6] and PaPi[7]. It was adapted from their open-sourced code, which contains the implementation of the techniques they used.
> > > >
> > > > - The `utils/util_loss.py` implemented how to integrate SAT loss into the baseline's loss function class.
> > > > - The warmup and mixup techniques are used by PiCO+ and PaPi. They utilized pseudo labels in their implementation for those. If we want to train the model with SAT loss and mixup loss, the mixup label should be the union of their partial label.
> > > >
> > > > If you are interested in the analysis of those designs, please refer to the original papers [5,6,7].

---

> > > > > ### Author Response · Authors · 2024-11-22
> > > > >
> > > > > ### **Q3:**
> > > > >
> > > > > >There is substantial research on the link between logical reasoning and machine learning, especially in areas like neuro-symbolic and abductive learning.
> > > > > >However, this work seems unrelated to logical reasoning, so the authors should not claim credit in that context.
> > > > >
> > > > > You are right that abductive learning and a lot of work in neuro-symbolic learning focus on the link between logical reasoning and machine learning. And you are also right that our work is unrelated to logical reasoning as we are working on PLL problem. Though they could be inspiring, the fields you mentioned are orthogonal to our work. We also did not claim credits regarding the contribution in those fields.
> > > > >
> > > > > >Furthermore, neuro-symbolic learning can be seen as a type of weakly supervised learning.
> > > > >
> > > > > The neuro-symbolic learning is obviously not a type of weakly-supervised learning. These two fields are barely connected. we can see that representative neuro-symbolic learning works [22,23] aim to either build a combinatorial solver to constraint the output of the neural network or construct the fuzzy logic elements into differentiable modules and integrate them into the end-to-end training of the neural network. They are totally different from our work (and general weakly supervised learning) regarding background, problems, methods, evaluation, etc.
> > > > >
> > > > > To the best of our knowledge, we are the first to bring logical constraints to PLL and conduct an extensive theory analysis of its merits to the problem. This is innovative for the PLL community and shows the potential to extend to other weakly-supervised learning problems, indicating our broad impact. Also, we never claim we created the idea to map logical constraints to loss functions. The contribution of our work mainly lies in how we cast the PLL problem, the theory analysis, and the experiment demonstration.
> > > > >
> > > > > Therefore, our claim is appropriate and justified.

---

> ### Comment · Reviewer_Ej8p · 2024-11-22
>
> Many thanks for your response and efforts in this rebuttal.
>
> 1. **About the property of SAT loss.**
>
> I agree with AC that the Rademacher complexity may not directly reflect whether a loss function is good or not.
> But the main concerns of mine persist.
> Why the SAT loss is better, and should be proposed?
> The author repeatedly claims that their method has *low entropy preference* and *monotonicity under ambiguity*.
> However, these two properties are quite trivial to construct as my initial review.
>
> - To the low entropy preference: it is easy to add a regularization loss on the entropy or just tune the temperature of the softmax.
> - To the monotonicity: As I said, this is mostly because of the $\prod$ operator, which is also very trivial. Moreover, this property can hinder the optimization process since the loss will be very small as the label size increases. The ad hoc way (as the authors enumerated) still faces the under-flow problem.
>
> I do not blame the author for the loss design being too simple, instead, I am very appreciate the simple way.
> However, after several rounds of rebuttals, I am sorry, that I still failed to capture the insight of this work.
>
>
> 2. **The code.**
>
> I have tried to reproduce some results based on the provided code to convince myself that this loss is better.
> However, I face challenges to do so, since there are many requirements and complex environmental issues.
> It would be better if the author provided a clean and easy-handle comparison code for SAT loss and other baselines.
> I am still working on this, so please be patient and calm down.
>
> 3. **The overclaim.**
> In fact, many researchers view neuro-symbolic learning as a special kind of weakly supervised learning. Name a few, [1,2].
> Also, we can easily apply the semantic loss to this task if you want, you just need to write the formula and call the semantic loss code. I do not mean the semantic loss is suited for this task, but as you can see, this is not very exciting.
>
>
>
> ---
> - [1] Zhi-Hua Zhou and Yu-Xuan Huang. Abductive Learning. In P. Hitzler and M. K. Sarker eds., Neuro-Symbolic Artificial Intelligence: The State of the Art, IOP Press, Amsterdam, 2022, p.353-379.
> - [2] Kaifu Wang, Efthymia Tsamoura, and Dan Roth. 2024. On learning latent models with multi-instance weak supervision. In Proceedings of the 37th International Conference on Neural Information Processing Systems (NIPS '23). Curran Associates Inc., Red Hook, NY, USA, Article 424, 9661–9694.
>
> ---
> I wish the author could benefit from the above discussions and revise their manuscript for a better quality.

---

> ### Comment · Reviewer_Ej8p · 2024-11-23
> **Some reproduced experiment result**
>
> ## Set up
>
> I have made the following changes:
>
> 1. Only keep the loss calculation part, and drop the regularization terms, e.g., mixup loss.
> 2. Run 500 epochs as the default option, since the discussion time is limited.
>
> Two methods are compared, one is SAT loss, as provided code; another is CC loss, which is copied from https://github.com/hongwei-wen/LW-loss-for-partial-label/blob/master/utils/utils_loss.py.
>
> The CC loss refers to the reference [12] and is a baseline compared in the paper.
>
> All experiments are running on the Ubuntu 24.04 noble with NVIDIA GeForce RTX 3090 and Intel Xeon Silver 4210 @ 40x 3.2GHz.
>
> For convenience, I put them in this block for easy comparison:
> ```py
>
> def cc_loss(outputs, partialY):
>     sm_outputs = F.softmax(outputs, dim=1)
>     final_outputs = sm_outputs * partialY
>     average_loss = -torch.log(final_outputs.sum(dim=1)).mean()
>     return average_loss
>
> def evaluate_disjunction_tensor(lineage:torch.tensor, softmax_predictions:torch.tensor)->torch.tensor:
>     # find all positions of 1's
>     indices = torch.nonzero(lineage, as_tuple=False)
>     # get the entries of those positions
>     predictions = softmax_predictions[indices]
>     # 1 - those entries
>     predictions =  1 - predictions
>     return 1 - torch.prod(predictions)
>
> def SAT_loss(outputs, targets, is_rel=None):
>     softmax_scores = torch.softmax(outputs, dim=1)
>     loss = torch.tensor([0.0], requires_grad = True).cuda()
>
>     if is_rel is None:
>         is_rel = torch.ones(outputs.shape[0], dtype=torch.bool).cuda()
>
>     for idx in range(softmax_scores.shape[0]):
>         if is_rel[idx]:
>             lineage = targets[idx,:]
>             loss_ = evaluate_disjunction_tensor(lineage, softmax_scores[idx,:]).cuda()
>             if loss_ > 9.2885e-30:
>                     loss = loss - torch.log(loss_)
>
>     loss = loss / is_rel.float().sum()
>
>     return loss
>
> ```
>
> ---
>
> ## Results
>
> The experiment was conducted on dataset cifar100 with a partial rate of 0.1.
>
> - SAT loss
> ```
> dataset:cifar100
>  partial rate:0.1
>  noisy rate:0
>  learning rate:0.02
>  COMMENT:None
> [2024-11-22 23:22:47]Epoch 0: Train Acc 1.96, Test Acc 3.64, Best Acc 0.00. (lr 0.0200,  output_mmc 0.03, L_total:2.31, loss_SAT:2.31, loss_cont:0.00)
> ...
> [2024-11-23 11:51:51]Epoch 240: Train Acc 79.92, Test Acc 66.17, Best Acc 66.51. (lr 0.0200,  output_mmc 0.85, L_total:0.37, loss_SAT:0.37, loss_cont:0.00)
> ...
> [2024-11-23 19:10:16]Epoch 499: Train Acc 84.54, Test Acc 66.51, Best Acc 66.95. (lr 0.0200,  output_mmc 0.91, L_total:0.23, loss_SAT:0.23, loss_cont:0.00)
>
> ```
>
> - CC loss
> ```
> dataset:cifar100
>  partial rate:0.1
>  noisy rate:0
>  learning rate:0.02
>  COMMENT:None
> [2024-11-23 09:55:03]Epoch 0: Train Acc 2.07, Test Acc 3.97, Best Acc 0.00. (lr 0.0200,  output_mmc 0.03, L_total:2.26, loss_CC:2.26, loss_cont:0.00)
> ...
> [2024-11-23 11:38:47]Epoch 240: Train Acc 79.99, Test Acc 66.09, Best Acc 66.74. (lr 0.0200,  output_mmc 0.85, L_total:0.35, loss_CC:0.35, loss_cont:0.00)
> ...
> [2024-11-23 13:30:21]Epoch 499: Train Acc 84.48, Test Acc 66.40, Best Acc 67.24. (lr 0.0200,  output_mmc 0.90, L_total:0.21, loss_CC:0.21, loss_cont:0.00)
> ```
>
> ---
>
> Two observations:
> 1. I confirmed that SAT loss can work well in this scenario but may not significantly outperform the baseline, at least in the compared setting.
> 2. I do let SAT loss run a longer time, as timestamps show, SAT loss is inefficient in computation.

---

> ### Author Response · Authors · 2024-11-29
>
> Dear Reviewer Ej8p,
>
> Thanks for taking the time to review our work again. However, it appears your implementation and result are inconsistent with our results and those reported in recent PLL literature published in top-tier conferences.
>
> Regarding SAT loss, as stated before, we applied the same training techniques as PiCO [5] for fair comparison. With your modifications to the code and different hyperparameters, it is not unexpected to see different results.
>
> For CC loss [12], you reported 66.74% for epoch 241 and 67.24% for epoch 500 without extra training tricks. We are confused as these numbers are higher than the results in recent PLL works (standard setting for CIFAR-100 with partial label rate = 0.1 and ResNet-18):
>
> |  Work |  Arch | Where | Epoch | CC Acc | Your Epoch | Your CC Acc |
> |--------|-------|---------|---------|----------|---------------|----------------|
> | [ICLR 2024 Oral: Candidate Label Set Pruning: A Data-centric Perspective for Deep Partial-label Learning](https://openreview.net/forum?id=Fk5IzauJ7F) | ResNet18 | Table 1 | 500+ | 62.31% | 500 | 67.24% |
> | [ICML 2022: Revisiting Consistency Regularization for Deep Partial Label Learning](https://proceedings.mlr.press/v162/wu22l.html) | ResNet34 | Table 2 | 300 | 58.65% | 241 | 66.74% |
> | [KDD 2022: Partial Label Learning with Semantic Label Representations](https://dl.acm.org/doi/10.1145/3534678.3539434) | ResNet18 | Table 1 | 300 | 56.39%| 241 | 66.74% |
>
> Other noisy PLL works also indicate that without extra training techniques, CC may not perform very well: [IJCAI 2023: Unreliable Partial Label Learning with Recursive Separation](https://dl.acm.org/doi/10.24963/ijcai.2023/468) and
>  [NeurIPS 2023: ALIM: Adjusting Label Importance Mechanism for Noisy Partial Label Learning](https://openreview.net/forum?id=PYSfn5xXEe&noteId=UF7iyTYQSN).
>
> Understandably, different implementations may result in differences in outcomes.
>
> We would like to thank you again for your effort and the time you spent reviewing and discussing our paper. However, the comparison you showed here misaligns with the reported results. We respect your stance on our work but strongly disagree with it.

---

### Official Review · Reviewer_A6aW · 2024-10-30

**Soundness:** 2
**Presentation:** 2
**Contribution:** 2
**Rating:** 3
**Confidence:** 4

**Summary:**

The paper proposes a novel loss function applicable to partial label learning, a problem in which each datapoint is annotated with a set of candidate labels, of which only one is the correct (or gold) one. They also provide the learning error bounds of this new loss and an extensive experimental analysis.

**Strengths:**

The paper is quite easy to follow.

**Weaknesses:**

The paper presents a series of dubious claims:

1. The authors first try to cast PLL as a satisfiability problem. This entails encoding the candidate set into a boolean formula which can be intuitively divided into two subformulas: one that states that at least one label in the candidate set should be the predicted and the other that states that only one should be predicted. Now, the authors can discard the second formula (which also happens to be very large) because they assume that they only work with models with softmax non-linearity in their last layer. This is fine, but then essentially their loss function is simply trying to maximise the probability that one of the labels in the candidate set should be predicted. There are many functions in the PLL community that try to do this (e.g., the methods based on the EM algorithm [1]). How is your loss different and better than these methods?

2. The authors call their loss the Saitifiability-based loss which from Remark 3.3 seems to simply coincide with the Product t-norm in the fuzzy logic literature. This is a failure to recognise the fact that the authors are *not* proposing a new loss, but they are applying a known loss function (which has been heavily studied in the Near-symbolic AI literature) in the PLL field.

3. The authors give a definition of the *small ambiguity degree*, however, that definition coincides with Cour's definition of ambiguity degree. If that is the small one, how would be large ambiguity degree be defined?


Smaller problems/typos:

- In Figure 2 what is PC, MC, PI and MI?
- Page 3 line 155 it should be the marginal of $P(X,S,Y)$
- Equation (2) it should be $X_{j'}$
- Corollary 4.2, of what exactly is a Corollary?

References:

[1] Jin, R., & Ghahramani, Z. (2002). Learning with multiple labels. In NeurIPS, Vol. 15 (pp. 897–904).

**Questions:**

See above

---

> ### Author Response · Authors · 2024-11-19
> **Author Rebuttal**
>
> Thanks for your review. We are happy to respond to resolve your concerns.
>
> ### **Q1:**
>
> Maximizing the probability that the predicted label to be in the candidate set is essentially the definition of partial label learning (or multiple-label learning) in the paper [2] you mentioned here. Of course, massive of PLL work tried to do so, as it is the purpose of the problem setting. We provided the theoretical analysis of why the specific math form of SAT loss can benefit that purpose. More importantly, the purpose of SAT loss is not only to maximize the probability that the predicted label is in the candidate set but also to maximize the prediction probability of one of the candidate labels (See Section 5 Theory Analysis - low entropy preference). Therefore, it benefits the label disambiguation
>
> We have conducted extensive experiments to demonstrate the superiority of SAT loss in mitigating the challenging scenario of high label ambiguity and fine-grained categorization. In addition, for the EM-algorithm-based method [2] you mentioned, we provide proof to show that CLPL[8] is a convex upper bound of SAT loss at the end of the rebuttal to Reviewer Ej8p. Based on proposition 6 in [8], The EM model is a convex upper bound of CLPL. Therefore, SAT loss is better than the previous EM-based methods you mentioned. As for the others, see Section 6 Experiment of our paper for details.
>
> ### **Q2:**
>
> The product t-norm, also known as the T-product or probabilistic product, is a specific operation [18] in fuzzy logic used to combine fuzzy sets. **In other words, product t-norm is NOT a loss function.** The term "product t-norm loss" is also not standard in mainstream machine learning literature, or more specifically Neuro-Symbolic AI community (if that is what you want to say). Therefore, it is unfair to say we are applying an existing loss.
>
> Remark 3.3 (now 4.3) expands on the conversion of the boolean formula in Example 4.1 into a loss function, as seen from the perspective of fuzzy logic literature. This enables the SAT loss to be interpreted through both logical and probabilistic lenses, bridging the two fields. As Reviewer 6sCy noted, this is a significant novelty of our work.
>
>
>
> ### **Q3: Difference between Small Ambiguity Degree and Ambiguity Degree.**
>
> If you check section 3.1 of Cour's paper[8], the description of the ambiguity degree under Equation (1), the ambiguity degree can vary in the range [0,1]. While Small Ambiguity Degree ranges in [0,1). That is to say, the Small Ambiguity Degree is a condition that there is no co-occurring label with the true label. If there exists a label that always appears in the candidate label set with the true label, there is no way to identify which is the ground truth. To the best of our knowledge, the concept/definition of the small ambiguity degree was first introduced in [10] and has also been used in many PLL works, such as [3,9]
>
> ### **Q4**
> We provided explanations based on the colors of the legend in both the diagram caption and the Introduction section. We will make it more explicit in the updated version. Thanks for your suggestion.
>
> PC: Prototype Correct
>
> MC: Model Correct
>
> PI: Prototype Incorrect
>
> MI: Model Incorrect
>
> ### **Q5 \& 6:**
>
> Thanks for the suggestion. We have corrected these typos.
>
> ### **Q7:**
> A corollary is a proposition that requires no or little proof based on an existing theorem or proposition. We use corollary here because the proof is comparatively easy. The usage of the term "corollary" can also be seen in [8].

---

> ### Comment · Reviewer_A6aW · 2024-11-21
>
> I thank the authors for their answer.
>
> However, in general I agree with the over claim stated by reviewer Ej8p.
>
> There are many works out there that map logical constraints to loss functions, and in particular that would map the constraints expressed by the authors in exactly the same way.
>
> The authors in their rebuttal write "The term "product t-norm loss" is also not standard in mainstream machine learning literature, or more specifically Neuro-Symbolic AI community (if that is what you want to say). "
>
> Available there is the LTN package [1] with implemented exactly the proposed loss which has 276 stars on GitHub: https://github.com/logictensornetworks/logictensornetworks. In other works this type of loss function has been called Semantic Based Regularization (SBR).
>
>
> [1] https://arxiv.org/abs/2012.13635

---

> ### Author Response · Authors · 2024-11-22
>
> Thanks for the reply. And we are happy to see the previous concerns have been well addressed.
>
>
> It is **untrue** to say that our work is overclaimed. Here is why.
>
> 1. To the best of our knowledge, we are the first to bring logical constraints to PLL and conduct an extensive theory analysis of its merits to the problem. This is innovative for the PLL community and shows the potential to extend to other weakly-supervised learning problems, indicating our broad impact. Also, we never claim we created the idea to map logical constraints to loss functions. The contribution of our work mainly lies in how we cast the PLL problem, the theory analysis, and the experiment demonstration.
>
> 2. The proposed SAT loss is built from the probability aspect. We provide Remark 4.3 to build a better literature connection with fuzzy logic for the readers. It provides another view to verify our proposed loss has good properties to apply logic constraints in the training of the neural network. In other words, the t-norm operation is neither the core of our proposed method nor a necessary step to construct our method.
>
> It is inappropriate to prohibit research work from using "Addition, subtraction, multiplication and division", just because these operations were invented before. The paper you provided, *Logic Tensor Networks*, is a good example to show it is generally acceptable even in the neuro-symbolic AI community to use t-norm in your work.
>
> 3. Our work is a Partial Label Learning work, instead of a neuro-symbolic one. There exist works trying to apply logical constraints for neural network training and they are doing entirely differently from ours.
>
> The GitHub repository you showed is the implementation of LTN, i.e. the paper you used to support your point, that they are doing the same thing as us. However, after a close look, it is apparently wrong.  The main method of the paper, Logic Tensor Networks, is the authors construct neural computational graphs from the elements of first-order logic to make those elements differentiable utilizing the proposed differentiable logical language, Real Logic. **They did not propose any new loss** but talked about how the loss function will be changed by the neural computational graphs.
>
> In addition, the term "Semantic Based Regularization" is not included in the paper you provided. This term comes from the paper [18] we cited in our Remark 4.3. However, Semantic Based Regularization targets the multi-objective problem, where the model has multiple optimization objectives. It is an entirely different domain compared to our work.
>
> 4. Finally, as a completion, we can see that representative neuro-symbolic learning works [22,23] aim to either build a combinatorial solver to constraint the output of the neural network or construct the fuzzy logic elements into differentiable modules and integrate them into the end-to-end training of the neural network. They are different from our work in terms of background, problems, methods, evaluation, etc. And we did not claim any contributions in those orthogonal fields.
>
> Therefore, our claim is appropriate and justified.

---

### Official Review · Reviewer_6sCy · 2024-11-06

**Soundness:** 4
**Presentation:** 4
**Contribution:** 3
**Rating:** 8
**Confidence:** 3

**Summary:**

A SAT-based formulation of the partial label learning problem is proposed. The authors encode each training instance (data / candidate-label pairs) into formulas in propositional logic. The proposed method then exploits a SAT-based loss to do training over these formulas. Theoretical justification is provided. The authors demonstrate (1.) robustness to label ambiguity- i.e. the risk a classifier suffers wrt the SAT loss is non-increasing with the partial label ambiguity. (2.) the SAT loss is proportional to the entropy of the network logits i.e., minimization of the SAT loss induces low-entropy confident predictions. The authors also provide a detailed analysis of the generalization of classifiers trained with the SAT loss.

Empirically, a significant improvement is demonstrated on a variety of datasets, including CIFAR100 and tiny-imagenet. Improved clustering of the latent space embeddings is demonstrated by applying T-SNE to model embeddings.

**Strengths:**

I find this paper to be very well-written and interesting- I appreciate the examples. The core concept is novel and I believe bridging neural methods and logical reasoning is an important topic of study. The inclusion of a detailed, but concise, theoretical analysis is beneficial in providing justification for their method. The empirical results are encouraging as well and I am happy to see that the authors have provided code. I am inclined to accept this paper.

**Weaknesses:**

see questions

**Questions:**

I assume there would be other methods that can be used to penalize the entropy of predictions. How does the proposed SAT loss compare to those methods?

If a subset of the partial labels are noisy how would SAT loss perform?

---

> ### Author Response · Authors · 2024-11-19
> **Author Rebuttal**
>
> Thank you very much for recognizing our work. We are happy to address your concerns.
>
> ### **Q1:**
>
> Exemplary methods that try to penalize the entropy of predictions could be [8] (mentioned by Reviewer Ej8p), [2] (mentioned by Reviewer A6aW) and [4,12] (our baseline).
>
> For [2,8], we provide proof to show that CLPL[8] and EM+Prior Model[2] provide the convex upper bound for our SAT loss, which means SAT loss is better. Full proof can be seen at the end of the rebuttal reply to Reviewer Ej8p.
>
> For [4,12], we have conducted extensive experiment results to demonstrate our method's superiority.
>
> Therefore, we showed our method is better than previous ones from both theory and practice aspects.
>
>
> ### **Q2:**
> We provided additional numerical results on the noisy partial label and real-world dataset in our Appendix B. The SAT loss still beats the crafted designed loss function methods and shows improvement when integrated with baselines with a complex training framework. For instance, SAT loss surpasses CC[12] loss by $10$\% on CIFAR-100 when there are $5$\% noisy labels. It improved the performance of $PiCO+$ [6] by ~$3$\% on CUB-200 when there are $20$\% noisy labels.

---

### Official Review · Reviewer_k18q · 2024-11-06

**Soundness:** 2
**Presentation:** 2
**Contribution:** 2
**Rating:** 5
**Confidence:** 4

**Summary:**

In this paper, the authors casted PLL into a satisfiability problem and incorporated a SATisfiability-based (SAT) loss based on this reduction. Besides, the authors proved several natural properties of satisfiability-based PLL: that the loss is non-increasing the larger candidate label sets become and that it favors low entropy classifiers. Experiments show that satisfiability-based training leads to consistently higher classification accuracy over all settings compared with previous state-of-the-art techniques.

**Strengths:**

The proposed SATisfiability-based (SAT) loss leads to consistently higher classification accuracy over all settings compared with previous state-of-the-art techniques.

**Weaknesses:**

1. **Lack of Intuitive Explanation for the Proposed Method's Effectiveness**: While the paper introduces the idea of "casting PLL into a satisfiability problem" to address performance drops in data with high label ambiguity and fine-grained categories, the intuition behind how this formulation resolves these issues is not sufficiently explained.

2. **Unclear Distinction Between PLL and Complementary-Label Learning**: The relationship between the proposed PLL method and Complementary-Label Learning (CLL) is not well articulated.

3. **Absence of Ablation Studies**: The paper lacks ablation studies that could provide more detailed insights into the contributions of each component of the proposed method. Additionally, while theoretical justifications for the SAT loss are provided, their direct relevance and contribution to the method’s practical effectiveness are not sufficiently explored.

**Questions:**

1. The approach of "casting PLL into a satisfiability problem" proposed in the paper is suggested to help address the issue of "significant performance drop on data with high label ambiguity and fine-grained categories." Could the authors provide some intuitive insights or explanations on how this formulation helps in this context? It appears that this approach could also address the issue of "low label ambiguity."

2. In the methodology section, the form of Eq. (3) appears similar to that used in Complementary-Label Learning. Based on this, it seems that the goal of minimizing the probability of labels not in the candidate label set is conceptually similar to the objectives in Complementary-Label Learning[1][2]. When the number of labels in the candidate label set increases, PLL seems to approach the problem of Complementary-Label Learning in some respects. Could the authors elaborate on the relationship between these two approaches?

3. In Section 4, the SAT loss is provided with theoretical justification. I would like to know whether these theoretical justifications offer any insights or guidance for the methodology and the underlying problem it addresses. For example, does Theorem 4.4 provide any valuable insights into the design of the proposed method?

4. The experiments lack corresponding ablation studies. Could the authors provide some potential ablation experiments or suggestions on which experiments could be conducted to further validate the proposed method?


[1] "Discriminative complementary-label learning with weighted loss." ICML 2021.

[2] "Learning with multiple complementary labels." ICML 2020.

---

> ### Author Response · Authors · 2024-11-19
> **Author Rebuttal**
>
> Thanks for your review. We are happy to respond for better clarity.
>
> ### **Explanation for Effectiveness**
> We provided explanations for why the proposed method is effective with theory justification in Section 5 THEORETICAL ANALYSIS. We would like to clarify it better combined with our motivation. For more details, please see section 5.
>
> - *Why do we use logic constraints to build such a formulation?*
>
> Previous methods rely on cross-entropy loss with pseudo labels for label disambiguation, which is proven to be unsatisfying when the PLL setting becomes challenging and close to real-world scenarios. Building a loss from logic constraints gets rid of the noisy pseudo-labels and other assumptions as a uniform prior distribution.
>
> - *Why SAT loss can benefit the high label ambiguity challenge?*
>
> As shown in Corollary 5.2, Monotonicity under ambiguity, the SAT loss is non-increasing along with the small ambiguity degree. In other words, the empirical risk (or expected loss over a dataset) of SAT loss is not affected by the increasing of ambiguity degree. Therefore, it shows its robustness to high label ambiguity.
>
> - *Why SAT loss can benefit fine-grained categorization?*
>
> Fine-grained categories mean the candidate labels could be very similar, such as Raven and Crow. This causes problems for cross entropy loss with the pseudo label as it becomes noisier. However, the SAT loss has a low entropy preference based on its logic-constraint nature. Specifically, we use the below example to illustrate this property.
>
> In a multi-class classification problem with classes A, B, C, and D, consider a data instance $x$ with a partial label $y = \\{0, 1, 0, 0\\}$, where B is the ground-truth label. For two predicted probability distributions:
> $$
> f_1(x) = \\{p_1^A = 0.25, p_1^B = 0.25, p_1^C = 0.25, p_1^D = 0.25\\}
> $$
> $$
> f_2(x) =\\{p_2^A = 0.01, p_2^B = 0.97, p_2^C = 0.01, p_2^D = 0.01\\},
> $$
> we calculate their corresponding SAT loss values as:
> $$
> l_{\text{SAT}}(f_1(x)) = 0.38, \quad l_{\text{SAT}}(f_2(x)) = 0.0295.
> $$
> This demonstrates that SAT loss encourages the model to concentrate predicted probabilities on a single label, thereby promoting label disambiguation.
>
>
> - *What does theorem 4.4 (now 5.4) tell us?*
>
> Theorem 5.4 in our paper provides a generalization bound for SAT loss. It builds the relationship between SAT loss and ideal 0/1 loss (optimal case), indicating the learnability and generalization ability of SAT loss.
>
> PS: We believe that integrating SAT loss shows improvement in "low label ambiguity" is a good signal, as it improves the robustness of the model towards high ambiguity and fine-grained categorization challenges but does affect the model's performance in such low ambiguity cases.
>
>
>
> ### **Relationship between PLL and CLL**
>
> In PLL, each training instance is associated with a set of candidate labels, one of which is the true label. The true label is hidden within this set. In CLL, instead of being given the correct label, the training instance is labeled with one or more complementary labels—labels that the instance does not belong to. This makes them essentially dual problems, both deal with the label ambiguity problem. CLL is a relatively new problem (first representative work [15] published in 2017) compared to PLL (first [2] seen in 2002).
>
>
> CLL could be seen as an extreme case of PLL from the label aspect. For most CLL works, there is no candidate label set but one single complementary label. However, these two domains target different optimization objectives and face different application scenarios. It is not a usual practice to compare methods from two different problems in weakly-supervised learning.
>
> Specifically, in our work, the target is different from CLL. CLL aims to minimize the prediction probability on the complementary labels. While our SAT loss aims to maximize the prediction probability of one of the candidate labels, which is achieved by its logic constraints.
>
> We appreciate you for raising this great question. We will definitely add the discussion of complementary label learning into our related work to complete the discussion.
>
> ### **Ablation Studies**
> Our experimental results naturally lead to ablation studies. We compared two categories of baseline methods. One is designed loss functions like SAT loss, which
> means they are different loss functions designed for PLL. The others are PLL frameworks like
> PiCO+[6], PaPi[7], and DPLL[4], which integrated multiple loss functions and additional learning
> modules such as representation prototypes, Momentum Contrast modules, etc.
>
> In Table 1, our main result, you can see the model performance trained with SAT loss only. Different baselines[4,6,7] performance with or without SAT loss. For instance, $PaPi$ and $PaPi_{SAT}$, SAT loss is integrated as a plug-in loss. Therefore, an additional table for the ablation study is a bit redundant.

---

### Official Review · Reviewer_58DH · 2024-11-08

**Soundness:** 3
**Presentation:** 3
**Contribution:** 2
**Rating:** 5
**Confidence:** 4

**Summary:**

This paper proposes a new method for Partial Label Learning (PLL) by formulating the task as a satisfiability (SAT) problem and introducing a SAT-based loss function. Partial label learning involves scenarios where each data instance has multiple candidate labels, but only one is correct. The authors argue that current PLL methods struggle in cases of high label ambiguity and fine-grained categorization. By encoding PLL constraints as logical formulas and designing a SAT loss that penalizes unsatisfied formulas, this method encourages disambiguation toward the true label without relying on pseudo-labels. The approach is validated on various benchmarks, showing up to 25% improvement over existing PLL techniques, such as PICO and PAPI, in certain settings.

**Strengths:**

Novel Problem Framing: Casting PLL as a SAT problem is innovative, providing a new angle to address label ambiguity. The SAT loss introduces a logic-driven constraint, which could contribute positively to PLL tasks by enforcing logical consistency.
Solid Theoretical Foundation: The paper includes theoretical analysis, demonstrating the robustness of SAT loss in high-ambiguity scenarios and fine-grained categorization tasks, supported by proofs of error bounds and properties like low-entropy preference.
Significant Experimental Improvements: Empirical results are extensive, covering diverse datasets and demonstrating significant performance gains. The experiments suggest that the SAT loss is effective in improving classification accuracy and resilience in challenging PLL scenarios.

**Weaknesses:**

Limited Innovation Scope in PLL: While the SAT-based formulation is novel, it may be perceived as incremental in its application, as it leverages established ideas from satisfiability and logical constraints. Whether this alone constitutes a breakthrough in PLL could be debatable, especially given the broader ICLR scope.
Dependency on Existing Models: The proposed SAT loss is integrated with existing PLL methods like PICO and PAPI rather than functioning independently as a standalone model. This limits the novelty of the overall algorithm, as the SAT loss relies heavily on baseline techniques for its effectiveness.
Complexity and Scalability Concerns: Implementing SAT loss with logic-based constraints may increase computational requirements. The scalability of the approach to large-scale, real-world datasets or cases with more complex dependencies among labels could be challenging and is not addressed in depth.

**Questions:**

Generalization to Real-World Data: Does the model maintain its efficacy when applied to real-world, noisy datasets where label ambiguity might not align as neatly with the SAT loss assumptions?

Computational Overheads: The SAT loss, while theoretically sound, could introduce computational complexity. How does this approach compare in terms of training efficiency with more traditional PLL methods?

Potential for Broader Application: Could the SAT-based formulation be generalized to other types of weakly supervised learning beyond PLL, such as multi-label learning?

Please clarify the significance of this partial label learning setting. Is it truly valuable to have numerous papers on this problem, given that each often presents only marginal improvements?

---

> ### Author Response · Authors · 2024-11-19
> **Author Rebuttal**
>
> Thanks for your review and suggestions. We are happy to respond for clarity.
>
> ### **Limited Scope in PLL**
>
> We aim to elaborate on our work from two perspectives:
>
> First, the **broad impact** of our work lies in its potential to inspire future research. Symbolism and connectionism have long been distinct paradigms, and while fields such as Neural-Symbolic AI are gradually attempting to unify them, leveraging symbolic reasoning to inspire deep learning remains at a very nascent stage. Our work represents a meaningful step in this direction, demonstrating that loss functions derived from logical constraints can provide additional benefits and desirable properties for neural network training. We have also provided theoretical proof to support this claim. Bridging logical constraints with neural methods is a core novelty of our work, as Reviewer 6sCy has noted. We believe this study can inspire not only the partial label learning (PLL) community but also the broader weakly supervised learning community, with promising extensions such as applications to multi-label learning, as you have suggested.
>
> Second, our work holds **practical significance** for real-world tasks, particularly for the broader PLL community. In many real-world scenarios, images often contain multiple objects, unlike the simplified single-object benchmarks commonly used in research. This highlights the practical relevance of PLL and its ability to address multi-object and multi-label learning challenges in real-world applications. Massive PLL works have been accepted in top-tier conferences such as ICML, NeurIPS, ICLR, CVPR, ECCV, and ICCV, which also demonstrated its value to the community. And their improvement is not marginal. As shown in [7], the classification accuracy under PLL on CIFAR-100 is only $0.08$% lower than the supervised learning (though with a very small ambiguity degree).
>
> Therefore, we believe that the scope of innovation in this work is not confined to PLL alone but has broader implications and impact.
>
> ### **Dependency on Existing Models**
>
> The SAT loss is not dependent on existing methods. We compared two categories of baseline methods. One is designed loss functions like SAT loss, which means they are just different loss functions designed for PLL. The others are PLL frameworks like PiCO+[6], PaPi[7], and DPLL[4], which integrated multiple loss functions and additional learning modules such as representation prototypes, Momentum Contrast modules, etc. Our experiment results give two points:
>
>
> 1. SAT loss can be used as a loss function alone for model training, and it surpassed the previous designed loss methods.
>
>
> 2. SAT loss can be smoothly integrated into existing PLL frameworks to improve their robustness to high-ambiguity and fine-grained categorization challenges. It can also be observed that integrating SAT loss can bring improvement even if the degree of ambiguity is small. These improvements do not depend on specific modules in existing methods, as [4,6,7] adopted different approaches to improve PLL.
>
>
> ### **Complexity and Scalability Concerns**
>
> For SAT loss, the primary computational cost arises from the product operation over the set of candidate labels $s$. **This operation is linear concerning the size of $s$, denoted as $|s|$.** The overall complexity per sample is $\mathcal{O}(|s|)$ for computing the product. Given that this needs to be done for every data point in a dataset of size $N$, the computational complexity becomes $\mathcal{O}(N|s|)$. Similarly, we can get the computational complexity for cross-entropy loss as $\mathcal{O}(NC)$, where $N$ is the number of the data samples and $C$ is the number of the classes ($C > |s|$).
>
> For a further discussion, as shown in [16], if the input propositional formula is both iDNF and 1OF (each variable occurs exactly once), then the computation of its probability (that is weighted model counting in our case) can be done by simply scanning the formula. Indeed as stated in Section 2 of that paper “In contrast, formulas in 1OF and iDNF admit probability computation in linear time.” (this text can be found in the paragraph under formula 3). Going back to our case, the propositional formula that expresses that either of the labels is the correct one is both iDNF and 1OF (excluding the mutual exclusiveness constraint).
>
> In practice, integrating SAT loss did not bring any noticeable extra training time. Therefore, the computational complexity of SAT loss is not a big concern, even for large-scale real-world datasets.

---

> > ### Author Response · Authors · 2024-11-19
> > **continue**
> >
> > ### **Generalization to Real-world Data**
> >
> > We provided additional numerical results in Appendix B, which is stated in our experiment section.
> >
> > The SAT loss maintains its efficacy under noisy labels and real-world data scenarios.
> >
> > For noisy label, SAT loss surpasses CC[12] loss by $10$% on CIFAR-100 when there are $5$\% noisy labels. It improved the performance of $PiCO+$ [6] by ~$3$\% on CUB-200 when there are $20$\% noisy labels.
> >
> > For real-world data, we chose Pascal VOC 2007 dataset [17], a widely used benchmark in visual object classification and detection tasks. The dataset is challenging due to the presence of multiple objects in varying scales, occlusions, and cluttered backgrounds. We showed that SAT loss improved the PaPi [7] by up to $16$\% on accuracy when $10$ out of the total $20$ classes are candidate labels. It brings ~$3$\% improvement for PiCO+[6], when there are just two candidate labels.
> >
> >
> > ### **Possible Extension to other weakly supervised learning problems**
> >
> > That’s a great question. First of all, as the terminology in the weakly-supervised learning literature is a bit overloaded, we would like to confirm the definition of multi-label learning (As [2] named PLL as multiple label learning). If the definition of multi-label learning here is that a data instance can have multiple labels that are all correct. Then the answer is yes.
> >
> > We could employ and extend the loss based on SAT loss. However, we have to make the following changes:
> >
> > The mutual exclusiveness formula, i.e., the subformula that states that only one label is the gold one, should be adopted to allow for non-exclusiveness.
> >
> > Discard the cross-entropy term, that is the negative logarithm, to prevent non-exclusiveness. We would need to somehow combine the binary cross entropy with the SAT loss.
> >
> > We believe this is a good example to show the broad impact of our work that can inspire future ideas that are not limited to the PLL community.

---

### Author Response · Authors · 2024-11-19
**Author Rebuttal**

Dear AC and reviewers,

We genuinely appreciate the valuable feedback from the reviewers on our work. We are encouraged by the reviewers' recognition that our work is well-written (Reviewer 6sCy, A6aW, Ej8p), with innovative motivations, interesting and promising
methods (Reviewer 58DH, 6sCy), a good combination of theoretical justification and superior experimental
performance (Reviewer 58DH, k18q, 6sCy, Ej8p).

We have responded to the concerns raised by reviewers in the individual replies. We hope these responses will offer additional clarity on the contributions and the novel approach taken in our work.


We appreciate your time and efforts in the review of our work. We are happy to discuss any further questions. It would be greatly appreciated if you could reconsider and raise the score!

We will use the below reference list in our replies:

[1] Suciu, D., Olteanu, D., Ré, C., & Koch, C. (2022). Probabilistic databases. Springer Nature.

[2] Jin, R., & Ghahramani, Z. (2002). Learning with multiple labels. Advances in neural information processing systems, 15.

[3] Cabannnes, V., Rudi, A., & Bach, F. (2020, November). Structured prediction with partial labeling through the infimum
loss. In International Conference on Machine Learning (pp. 1230-1239). PMLR.

[4] Wu, D. D., Wang, D. B., & Zhang, M. L. (2022, June). Revisiting consistency regularization for deep partial label learning.
In International conference on machine learning (pp. 24212-24225). PMLR.

[5] Wang, H., Xiao, R., Li, Y., Feng, L., Niu, G., Chen, G., & Zhao, J. (2022). Pico: Contrastive label disambiguation for partial
label learning. In International conference on learning representations.

[6] Wang, H., Xiao, R., Li, Y., Feng, L., Niu, G., Chen, G., & Zhao, J. (2023). Pico+: Contrastive label disambiguation for robust
partial label learning. IEEE Transactions on Pattern Analysis and Machine Intelligence.

[7] Xia, S., Lv, J., Xu, N., Niu, G., & Geng, X. (2023). Towards effective visual representations for partial-label learning. In
Proceedings of the IEEE/CVF Conference on Computer Vision and Pattern Recognition (pp. 15589-15598).

[8] Cour, T., Sapp, B., & Taskar, B. (2011). Learning from partial labels. The Journal of Machine Learning Research, 12, 1501-
1536.

[9] Lv, J., Xu, M., Feng, L., Niu, G., Geng, X., & Sugiyama, M. (2020, November). Progressive identification of true labels for partial-label learning. In international conference on machine learning (pp. 6500-6510). PMLR.

[10] Liu, L., & Dietterich, T. (2014, June). Learnability of the superset label learning problem. In International conference on machine learning (pp. 1629-1637). PMLR.

[11] Wen, H., Cui, J., Hang, H., Liu, J., Wang, Y., & Lin, Z. (2021, July). Leveraged weighted loss for partial label learning. In
International conference on machine learning (pp. 11091-11100). PMLR.

[12] Feng, L., Lv, J., Han, B., Xu, M., Niu, G., Geng, X., ... & Sugiyama, M. (2020). Provably consistent partial-label learning.
Advances in neural information processing systems, 33, 10948-10960.

[13] Feng, L., Kaneko, T., Han, B., Niu, G., An, B., & Sugiyama, M. (2020, November). Learning with multiple complementary
labels. In International conference on machine learning (pp. 3072-3081). PMLR.

[14] Mohri, M. (2018). Foundations of machine learning.


[15] Ishida, T., Niu, G., Hu, W., & Sugiyama, M. (2017). Learning from complementary labels. Advances in neural information processing systems, 30.

[16] Fink, R., Huang, J., & Olteanu, D. (2013). Anytime approximation in probabilistic databases. The VLDB journal, 22, 823-848.

[17] Everingham, M. (2009). The PASCAL visual object classes challenge 2007. In http://www. pascal-network. org/challenges/VOC/voc2007/workshop/index. HTML.

[18] Diligenti, M., Gori, M., & Sacca, C. (2017). Semantic-based regularization for learning and inference. Artificial Intelligence, 244, 143-165.

[19] Thomson, B. S., Bruckner, J. B., & Bruckner, A. M. (2008). Elementary real analysis (Vol. 1). ClassicalRealAnalysis. com.

[20] O'Searcoid, M. (2006). Metric spaces. Springer Science & Business Media.

[21] Kakade, S. M., Sridharan, K., & Tewari, A. (2008). On the complexity of linear prediction: Risk bounds, margin bounds, and regularization. Advances in neural information processing systems, 21.

[22] Paulus, A., Rolínek, M., Musil, V., Amos, B., & Martius, G. (2021, July). Comboptnet: Fit the right np-hard problem by learning integer programming constraints. In International Conference on Machine Learning (pp. 8443-8453). PMLR.

[23] Pogančić, M. V., Paulus, A., Musil, V., Martius, G., & Rolinek, M. (2020). Differentiation of blackbox combinatorial solvers. In International Conference on Learning Representations.

---

> ### Author Response · Authors · 2024-11-20
> **Update on Revision 1**
>
> Dear AC and Reviewer,
>
> We would like to kindly let you know that we uploaded a revision of the paper. The major changes are listed below:
>
> 1. We moved the Related Work Section from section 6 to section 2 and merged or adjusted several paragraphs for a better layout.
>
> 2. We corrected the typos and wording based on the reviewers' suggestions.
>
> There are no changes regarding the content of the paper's main body. We provided the latest section index in our rebuttal.
>
> Best,
> Authors.

---

### Comment · Area_Chair_KsDr · 2024-11-22
**calm down**

Please calm down and stop your discussion, because you cannot get the points of each other effectively. You are just repeating your original statements to the air and this would not change anything in the end.

---

> ### Comment · Area_Chair_KsDr · 2024-11-22
>
> Reviewer Ej8p, thanks very much for your service so far!
>
> In a reply to you (https://openreview.net/forum?id=Z25xTjf3Mv&noteId=8Y7ZfTWZ7v), the authors have shown that `pseudo_labels_b` is computed but not used later (at least in the part shown in the reply). I think this is understandable, because they downloaded and modified some existing codes, possibly without cleaning up all unnecessary lines.
>
> So have you found other places involving the pseudo labels? If not, let us regard the proposal not using the obtained pseudo labels.

---

> ### Comment · Area_Chair_KsDr · 2024-11-22
>
> Authors, thanks for your contribution to ICLR so far.
>
> In another reply (https://openreview.net/forum?id=Z25xTjf3Mv&noteId=JpCxuHEMTu), I don't think you get the point. The reviewer didn't mean simply rescaling the same loss function itself and he or she repeated this point a few times. So let me give you some examples.
>
> Suppose that I have a loss function named BCE2 as 1.6 times BCE. As a result, this new loss is an upper bound of the hinge loss (and they are very close to each other). Obviously, for training a linear-in-parameter model that leads to convex optimizations for the above losses, BCE2 is not better than hinge and hinge is not better than BCE2 either.
>
> Indeed, since BCE is sqrt(2) times the original logistic loss, I can always construct a strange loss function lying between logistic and BCE and being worse than them when used in training. Given that this loss is a strictly tighter loss than BCE and logistic is a strictly tighter loss than this loss, we can see that tightness itself doesn't imply anything meaningful.
>
> If we allow non-convex optimizations (which is always the case nowadays), the ramp loss is a lower bound of the hinge loss and the 0-1 loss is a lower bound of the ramp loss, and the corresponding inequalities are tight. No loss can be tighter than 0-1 loss itself as an upper bound of 0-1 loss, but you never want to optimize 0-1 loss in practice.
>
> Therefore, "it is crucial to analyze whether the loss function contributes to smoother training, reduces variance, and lowers computational cost".

---

> ### Comment · Area_Chair_KsDr · 2024-11-22
>
> Considering the Rademacher complexity debate, I don't think it's a major issue because I don't think it's an important part. Isn't it just a decoration for the submission?
>
> That being said, I'd like to give my two cents. Let us forget the looseness of the Rademacher complexity of function classes and the symmetrization technique used to bound the error using the Rademacher complexity. My point is that all contraction lemmas for Rademacher averages are loose! Basically, all of them are based on the original proof in the book "Probability in Banach Spaces: Isoperimetry and Processes" which you may be interested in to check.
>
> All contraction lemmas need to involve the Lips const of the function composed with the given function class to obtain an upper bound of the Rademacher complexity of the composed function class. Why Lips const? It's the worse-case smoothness measure, not the average-case smoothness measure.
>
> For the squared loss or anything between the MAE and MSE, it's not Lips continuous and thus the Lips const is infinity. Even if we limit the range where yf(x) can take the value, the Lips const of the squared loss in this range can still be extremely large. However, the squared loss is good. When yf(x) is far away from 1, its large local Lips const makes optimization quickly go to the smaller-objective region, and when yf(x) is close to 1, the local Lips const is actually even smaller than 1 making optimization here even more stable than the case using the MAE loss. The MAE loss has much better smoothness than MSE (both worst- and average-cases), but MSE usually outperform MAE if pre-processing is done well.
>
> Therefore, providing a bound based on the Rademacher complexity is not a problem at all, but I don't see further implications the authors want to convey to the readers besides the decorating purpose. I'm not blaming the authors; this has been a common issue for machine learning papers for many years.

---

> ### Author Response · Authors · 2024-11-23
> **Author Reply to "Tightness" Debate**
>
> Dear AC,
>
> Thank you for giving more examples. We understand that "tightness" alone can not imply one loss is better than another loss. And we agree the core of the discussion is not about "tightness" but the advantages of our method. We give the below discussion for better support of our view.
>
>
> **SAT loss is better in experiments.**
>
> For the work [8,12] mentioned by Reviewer Ej8p, CLPL[8] is one of the baseline methods of [12], which has been shown to perform much worse than [12] in their experiments. Recent PLL works[4,5,6,7,11] published in top-tier conferences have already stopped using CLPL[8] as a baseline as it is a classic early work. [12], RC and CC, is one of our baseline methods. As shown in our experiment section, we have demonstrated the proposed SAT loss has better performance over multiple challenging datasets and different partial label settings.
>
>
> **We have provided classic loss comparison.**
>
> CLPL[8] is one of the most classic and important PLL works. When compared with the previous method [2], the authors provided Surrogate Loss Analysis, Generalization Bounds, and Experiment Results to show their loss is better. We have provided the same analysis in our paper and rebuttal.
>
> 1. Surrogate Loss Analysis
>
> - upper bound relationship between SAT loss and CLPL. [\[Rebuttal Reply\]](https://openreview.net/forum?id=Z25xTjf3Mv&noteId=U1T9WUN8In)
>
> $$
> \ell_{SAT} \leq \ell^{\text{max}}_{\psi}, \quad \text{for all x}
> $$
>
> - Proposition 6 in [8]
>
> $$
> 2 \ell^P_{0/1} \leq \ell_\psi^{\text{max}} \leq \ell_\psi \leq \ell_\psi^{\text{naive}}
> $$
>
> where $\ell^P_{0/1}$ is the 0/1 loss in PLL; $\ell^{\text{naive}}_\psi$ is the method [2].
>
> - Lemma A.5 in our appendix
> $$
> \ell^P_{0/1} \leq \ell_{SAT}
> $$
>
> Combining the above, we can easily extend proposition 6 (surrogate loss analysis) by including $\ell_{SAT}$, resulting:
>
> $$
> \ell^P_{0/1} \leq \ell_{SAT} \leq \ell_{\psi}^{\text{max}} \leq \ell_{\psi} \leq \ell_\psi^{\text{naive}}
> $$
>
> 2. Generalization Bounds
>
> We have provided generalization bounds in our paper, Theorem 5.4.
>
>
> 3. Experiment Results
>
> We have discussed above.
>
> Therefore, we have provided a discussion from theory and experiment to show our method is advantageous (may not be perfect but a known good practice based on accepted works).
>
> **SAT loss has good behavior during training.**
>
> We can compare the loss curve, accuracy curve, and model prediction confidence curve between Cross-Entropy Loss (PaPi) and SAT loss (PaPi + SAT by replacing cross-entropy loss).
>
> [[Image Link]](https://anonymous.4open.science/r/Rebuttal_image-9A1B/rebuttal_image.pdf)
>
> It can be observed that:
>
> 1. In Figure 1 loss curve, the SAT loss converges faster and better. It quickly approached 0 with just 200 epochs. It results in better convergence at the end compared to CE loss. We can also observe the SAT loss has a much smaller loss value from the beginning, because of the *Monotonicity under ambiguity* property.
>
> 2. In Figure 2 accuracy loss, the accuracy of the SAT method grows faster along with the loss declining. The SAT loss reached $40$\% accuracy within 50 epochs while cross-entropy loss spent over 400 epochs.
>
> 3. In Figure 3 model prediction confidence curve, we show the change of highest prediction probability among all the labels (average over all the data). The model prediction confidence is a good indicator to show the disambiguation process. We can see SAT loss increased model prediction confidence much faster than cross-entropy. In the end, it has a higher model confidence close to 1 than cross-entropy. This shows how *Low entropy preference* benefits label disambiguation.
>
> We hope the above explanation can provide better insights into our methods and address the concerns of the Reviewers.

---

> ### Author Response · Authors · 2024-11-23
> **Author Reply to Rademacher complexity debate**
>
> Thank AC for clarifying that directly comparing the Rademacher complexity between loss functions is still an open problem. We hope this can address the initial concern of Reviewer Ej8p regarding this.
>
> In addition, we would like to add the comment that using Rademacher complexity to build the generalization bounds is an important part of many PLL works accepted by top-tier conferences/Journals [8,10,11,12]. We also provided such analysis (Theorem 5.4) and a non-trivial proof.

---

> > ### Comment · Area_Chair_KsDr · 2024-11-23
> >
> > Hi authors,
> >
> > BTW, I found that you didn't cite the first PLL paper, "Learning with Multiple Labels" by Rong Jin and Zoubin Ghahramani at NeurIPS 2002. This paper is actually about partial-label learning rather than multi-label learning or crowdsourcing.
> >
> > You should definitely look at it! Eq. (1) and Eq. (5) are the origins of the average-based strategy and identification-based strategy!
> >
> > AC

---

### Note · Authors · 2024-11-30

**Comment:**

Unfortunately, despite our efforts to address the majority of concerns, we were still unable to reach a consensus with some reviewers regarding their feedback.

**Withdrawal Confirmation:**

I have read and agree with the venue's withdrawal policy on behalf of myself and my co-authors.